# Zero-Shot Sharpness-Aware Quantization for Pre-trained Language Models

**Miaoxi Zhu**[1]*, **Qihuang Zhong**[1]*, **Li Shen**[2], **Liang Ding**[3], **Juhua Liu**[1]†
**Bo Du**[1]†, **Dacheng Tao**[3]

[1] School of Computer Science, National Engineering Research Center for Multimedia Software, Institute of Artificial Intelligence and Hubei Key Laboratory of Multimedia and Network Communication Engineering, Wuhan University, China

[2] JD Explore Academy, China    [3] The University of Sydney, Australia

{zhumx,zhongqihuang,liujuhua,dubo}@whu.edu.cn, {mathshenli,liangding.liam,dacheng.tao}@gmail.com

## Abstract

Quantization is a promising approach for reducing memory overhead and accelerating inference, especially in large pre-trained language model (PLM) scenarios. While having no access to original training data due to security and privacy concerns has emerged the demand for zero-shot quantization. Most of the cutting-edge zero-shot quantization methods primarily ❶ apply to computer vision tasks, and ❷ neglect of overfitting problem in the generative adversarial learning process, leading to sub-optimal performance. Motivated by this, we propose a novel zero-shot sharpness-aware quantization (ZSAQ) framework for the zero-shot quantization of various PLMs. The key algorithm in solving ZSAQ is the SAM-SGA optimization, which aims to improve the quantization accuracy and model generalization via optimizing a minimax problem. We theoretically prove the convergence rate for the minimax optimization problem and this result can be applied to other nonconvex-PL minimax optimization frameworks. Extensive experiments on 11 tasks demonstrate that our method brings consistent and significant performance gains on both discriminative and generative PLMs, *i.e.*, up to +6.98 average score. Furthermore, we empirically validate that our method can effectively improve the model generalization.

## 1 Introduction

Pre-trained language models (PLMs), such as BERT (Devlin et al., 2018) and GPT-3 (Brown et al., 2020), have achieved great success in a variety of NLP tasks (Raffel et al., 2020; Zhong et al., 2022b, 2023a; Liu et al., 2023b). However, with the scaling of model size, the inference of larger PLMs, *e.g.*, OPT (Zhang et al., 2022) and LLaMA (Touvron et al., 2023), becomes more computationally expensive and memory-intensive (Shen et al.,

2023). Hence, it is crucial and green to lighten the PLMs and reduce the memory footprint (Schwartz et al., 2020).

To achieve this goal, various model compression methods have been developed, including knowledge distillation (Liu et al., 2019), pruning (Liu et al., 2018) and *etc*. Among these methods, quantization has attracted great attention (especially in the large language model scenarios) (Yao et al., 2022; Frantar et al., 2022), owing to its impressive ability to reduce memory footprint and accelerate inference while maintaining the network structure (Bai et al., 2021a). However, quantization-aware training (QAT) usually requires retraining using the original training data to mitigate performance degradation. While in some application scenarios, access to the original training data is oftentimes not available due to the protection of privacy and consideration of data security.

In response to this problem, post-training quantization (PTQ) (Bai et al., 2021a) is proposed to address the training-data access, as it generally requires no retraining. But most of those methods would lead to an accuracy drop in low-precision quantization since they are training-free and primarily rely on the analysis of weight distributions. More recently, zero-shot quantization (ZSQ) (Nagel et al., 2019; Zhuang et al., 2022; Zhang et al., 2021) shows promising results in the computer vision community. Specifically, ZSQ performs the quantization process with the synthetic data generated from a generator. To alleviate the side effect of inaccurate synthetic data, Choi et al. (2020) further improve the ZSQ by adaptively tuning the generator with the supervision of adversarial learning. However, it is non-trivial to adopt such an adversarial-based ZSQ in the NLP field, as ❶ backpropagation on discrete words is not reasonable, *i.e.*, it seems "unlikely" to pass the gradient through the text to the generator; ❷ minimizing the difference between the teacher and (quantized) stu-

---

*Miaoxi Zhu and Qihuang Zhong contribute equally to this work.

† Corresponding Authors: Juhua Liu (e-mail: liujuhua@whu.edu.cn), Bo Du (e-mail: dubo@whu.edu.cn)

dent models would be prone to over-fitting problem, leading to poor model generalization.

To address the above issues, we propose a novel Zero-shot Sharpness-Aware Quantization (namely ZSAQ) framework to improve the performance and generalization of the quantized model. Firstly, regarding the problem ❶, we design a simple feature adaptation module to convert the output token representations of the generator into the target quantized model's representation space, thus avoiding the gradient propagation on the discrete words. Secondly, for the problem ❷, we are inspired by the sharpness-aware minimization (SAM) (Foret et al., 2020) and propose an alternating SAM-SGA algorithm to boost the ZSAQ and improve model generalization. Specifically, on the one hand, SAM-SGA uses SAM to robustly minimize the divergence of output distributions between teacher and quantized student models. On the other hand, it uses stochastic gradient ascent (SGA) to encourage the generator model to maximize the divergence at each iteration. By optimizing such a minimax problem, we can not only train the well-performed with generative adversarial learning but also improve the generalization of quantized models.

Theoretically, we provide rigorous analysis for the SAM-SGA algorithm solving the minimax optimization (ZSAQ). We show that SAM-SGA achieves $\mathcal{O}(1/\sqrt{T})$ convergence rate. Furthermore, our theoretical results are not limited to the single case but are applicable to a wide range of minimax optimization problems in both nonconvex-strongly-concave and nonconvex-PL conditions. Empirically, we adopt our ZSAQ framework to quantize both the discriminative and generative PLMs, and evaluate 8 language understanding tasks and 3 language modeling tasks. Extensive experiments demonstrate the effectiveness and versatility of our approach. More encouragingly, our ZSAQ brings +6.98 average performance gains in the low-precision quantization, compared to the baselines. Extensive analyses show that our framework has the potential to expand to more large language models and prove that SAM-SGA indeed brings better model generalization.

In summary, our contributions are three-fold: (1) We propose a novel method, zero-shot sharpness-aware quantization (ZSAQ), which realizes zero-shot quantization without much accuracy loss. (2) We provide a theoretical convergence guarantee for the SAM-SGA algorithm which aims to solve the minimax optimization problem (ZSAQ). (3) Extensive experiments show that our SAM-SGA can bring consistent and significant performance gains, up to +6.98 average score, on both discriminative and generative PLMs.

## 2 Related Works

**Compression method for language models.** Compression is efficient in reducing computation cost, memory overhead, and energy consumption as well as shortens inference time for large neural network models, which is in great need for pre-trained language models to be deployed in the mobile or resource-constrained device. Quantization works by transmitting the models with lower-bit parameters. QAT works by minimizing the rounding error on the training dataset. FullyQT (Prato et al., 2020) shows the fully quantized Transformer can avoid any accuracy loss in translation quality. I-BERT (Kim et al., 2021) eliminates floating point calculation in BERT inference and achieves similar accuracy to the full-precision baseline. BinaryBERT (Bai et al., 2021b) achieves good performance by ternary weight splitting which avoids the complex and irregular loss landscape. There are some other techniques, low-rank factorization (Tahaei et al., 2021; Edalati et al., 2021; CHen et al., 2020; Reid et al., 2021), factorizing the weight matrices which are usually low-rank into several smaller matrices by means of singular value decomposition. Parameter sharing (Rothe et al., 2020; Takase and Kiyono, 2021; Reid et al., 2021), reduces memory overhead by reusing the same parameters in multiple computations. Pruning (Mishra et al., 2021; Guo et al., 2020; Chen et al., 2020; Xia et al., 2022; He et al., 2022; Liu et al., 2023a), deletes some parameters which are "useless" in network under the same model performance.

**Zero-shot quantization.** In contrast to QAT, PTQ relies less on the training data and does not require end-to-end training (Nagel et al., 2019; Nahshan et al., 2021; Nagel et al., 2020; Li et al., 2020; Zhao et al., 2019; Hubara et al., 2020), which can be zero-shot. Bai et al. (2021a) propose module-wise quantization error minimization and perform close to QAT. SmoothQuant (Xiao et al., 2022) transmits the activation quantization to weights to smooth the outliers in activation and makes a large language model implemented efficiently. Tao et al. (2022) proposes an adaptive quantization for different modules and performs compara-

bly with the full-precision generative pre-trained language models. GPTQ (Frantar et al., 2022), a one-shot weight quantization method with approximate second-order information, can efficiently quantize generative pre-trained transformers with large amounts of parameters to 3 or 4 bits with negligible accuracy degradation. Nahshan et al. (2021) figures out that aggressive quantization can lead to the steep curvature of the loss landscape, while they propose a method combining layer-by-layer quantization and multivariate quadratic optimization that can improve accuracy over current PTQ. NuQmm (Park et al., 2022) uses a non-uniform quantization method which allows for a trade-off between accuracy and compression ratio. Another way to achieve zero-shot is by means of the generative adversarial technique. (Cai et al., 2020; Liu et al., 2021b; Choi et al., 2022) has shown promising performance on small-scale computer vision tasks, while it has not been applicable to the NLP realm. To the best of our knowledge, we are the first to adopt the adversarial learning approach for quantizing both the discriminative and generative PLMs in a zero-shot manner and provide its rigorous convergence analysis.

## 3 Methodology

In this section, we will present our zero-shot sharpness-aware quantization method.

**Network quantization.** For a real-valued vector $\mathbf{x}$ which represents the parameter in the pre-trained full-precision model, we quantize it into $\hat{\mathbf{x}}$ by the signed symmetric quantization:

$$\hat{\mathbf{x}} = s \left[ clamp \left( \lfloor \frac{\mathbf{x}}{s} \rceil; -2^{b-1}; 2^{b-1} - 1 \right) \right], \quad (1)$$

where $s \in \mathbb{R}^+$ denotes scale factor and $b \in \mathbb{Z}$ represents the bit-width of the quantized version. So the parameters can be projected to the integer grid: $\{-2^{b-1}, -2^{b-1} + 1, ..., 0, ..., 2^{b-1} - 1\}$. And the default experimental setting is presented in Sec. 5.

**Generative adversarial learning.** In our paper we achieve zero-shot by generative adversarial learning. Specifically, the generator $\mathcal{G}$ takes i.i.d. token samples $z$ as input to produce synthetic data. Then the generated sentence segments will be fed into the pre-trained model $\mathcal{P}$ and the quantized model $\mathcal{Q}$. Intuitively we can get the output distributions, denoted as $\mathcal{P}(\mathcal{G}(z))$ and $\mathcal{Q}(\mathcal{G}(z))$ respectively.

During the training process of the quantized model, it aims to simulate the output distribution

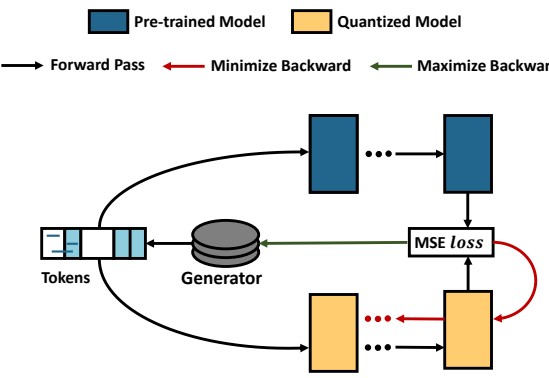

Figure 1: An overview of our ZSAQ framework.

of the pre-trained model. So the objective for $\mathcal{Q}$ is to minimize the discrepancy between $\mathcal{P}(\mathcal{G}(z))$ and $\mathcal{Q}(\mathcal{G}(z))$, denoted as $\mathcal{D}(\mathcal{P}(\mathcal{G}(z)), \mathcal{Q}(\mathcal{G}(z)))$. While for the generator $\mathcal{G}$, it tries to maximize the divergence, so that the quantized model $\mathcal{Q}$ can be trained to confuse even the most strict generator. We try to balance the trade-off between the generator $\mathcal{G}$ and the quantized model $\mathcal{Q}$ so that we can get a well-performed quantized model without access to the original training data. The structure of our method is demonstrated in the Figure 1. And the objective function of our method is shown below:

$$\min_{\mathcal{Q}} \max_{\mathcal{G}} \mathbb{E}_{z \sim \mathcal{T}} \mathcal{D}(\mathcal{P}(\mathcal{G}(z)), \mathcal{Q}(\mathcal{G}(z))), \quad (2)$$

where $\mathcal{D}$ represents the divergence MSE.

**ZSAQ: Zero-shot sharpness-aware quantization.** As shown in Liu et al. (2021a), there is a sharper loss landscape in the low-precision model than in the full-precision model. To avoid overfitting problems caused by the training process, we should consider the generalization ability of our quantized models. Inspired by Foret et al. (2020), we pursue a flatter landscape during the training process of the quantized model which can guarantee a better generalization performance. So we can adjust our objective function as:

$$\min_{\hat{\omega}} \{ \max_{\mathcal{G}} \{ \mathbb{E}_{z \sim \mathcal{T}} \mathcal{D}(\mathcal{P}(\mathcal{G}(\mathbf{z})), \mathcal{Q}(\hat{\omega}; \mathcal{G}(\mathbf{z}))) \} + \mathcal{S}(\mathcal{P}; \mathcal{Q}(\hat{\omega})) \}, \quad (3)$$

where we denote the sharpness function as:

$$\mathcal{S}(\mathcal{P}; \mathcal{Q}(\hat{\omega})) \quad (4)$$
$$\triangleq \max_{\|\epsilon\|_2 \leq \beta} \mathbb{E}_{z \sim \mathcal{T}} \mathcal{D}(\mathcal{P}(\mathcal{G}(\mathbf{z})), \mathcal{Q}(\hat{\omega} + \epsilon; \mathcal{G}(\mathbf{z})))$$
$$- \mathbb{E}_{\mathbf{z} \sim \mathcal{T}} \mathcal{D}(\mathcal{P}(\mathcal{G}(\mathbf{z})), \mathcal{Q}(\hat{\omega}; \mathcal{G}(\mathbf{z}))) \}.$$

The sharpness function searches for the sharpest point among a neighbor centered at $\hat{\boldsymbol{\omega}}$ with a radius of $\beta$, which will represent the worst-case data. Then our optimization process not only minimizes the divergence but also the sharpness of the divergence, i.e., getting a quantized model with both lower divergence and a flatter landscape.

Solving the inner maximization problem to obtain the optimal:

$$\epsilon^*(\hat{\boldsymbol{\omega}}) = \beta \frac{\nabla_{\boldsymbol{\omega}} \mathbb{E}_{z \sim \mathcal{T}} \mathcal{D}(\mathcal{P}(\mathcal{G}(\mathbf{z})), \mathcal{Q}(\hat{\boldsymbol{\omega}}; \mathcal{G}(\mathbf{z})))}{\|\nabla_{\boldsymbol{\omega}} \mathbb{E}_{z \sim \mathcal{T}} \mathcal{D}(\mathcal{P}(\mathcal{G}(\mathbf{z})), \mathcal{Q}(\hat{\boldsymbol{\omega}}; \mathcal{G}(\mathbf{z})))\|_2}. \quad (5)$$

**Remark.** Here we use $\hat{\boldsymbol{\omega}} + \epsilon$ as the parameter in the sharpness function (4) instead of $\widehat{\boldsymbol{\omega} + \epsilon}$, for the reason that $\epsilon$ may be so small that there will be no difference adding the perturbation. And this issue is also illustrated in Liu et al. (2021a).

**Algorithm.** For the objective function, we train the generator to maximize the discrepancy between the pre-trained model and the quantized model:

$$\mathbb{E}_{z \sim \mathcal{T}} \mathcal{D}(\mathcal{P}(\mathcal{G}(\mathbf{z})), \mathcal{Q}(\hat{\boldsymbol{\omega}}; \mathcal{G}(\mathbf{z}))) \quad (6)$$

Then we train the quantized model to minimize both the discrepancy value and the sharpness of the discrepancy, i.e., the addition of $\mathcal{D}(\mathcal{P}, \mathcal{Q})$ and $\mathcal{S}(\mathcal{P}, \mathcal{Q})$, on the synthetic data produced by the generator $\mathcal{G}$. While the addition comes out to be:

$$\mathbb{E}_{z \sim \mathcal{T}} \mathcal{D}(\mathcal{P}(\mathcal{G}(\mathbf{z})), \mathcal{Q}(\hat{\boldsymbol{\omega}} + \epsilon^*(\hat{\boldsymbol{\omega}}); \mathcal{G}(\mathbf{z})))\}, \quad (7)$$

where the optimal $\epsilon^*(\hat{\boldsymbol{\omega}})$ is obtained from Eq. (5).

We obtain our final algorithm by applying stochastic gradient descent on the minimization objective (7), which turns out to be SAM, and applying stochastic gradient ascent on the maximization objective (6). Generator $\mathcal{G}$ is parameterized by $\boldsymbol{\theta}$, quantized model $\mathcal{Q}$ initially quantized from the pre-trained model $\mathcal{P}$ and the quantized parameter is denoted as $\hat{\boldsymbol{\omega}}$, while the parameters before quantization is denoted as $\boldsymbol{\omega}$. We alternatively update the parameters $\hat{\boldsymbol{\omega}}$ and $\boldsymbol{\theta}$ respectively. The pseudo code is shown in the Algorithm 1.

## 4 Optimization

In this part, we will analyse the convergence of our algorithm to solve this minimax optimization problem, which can provide a theoretical guarantee for our method.

First in order to make our theoretical convergence result applicable to a broader variety of such problems, we replace the detailed loss function

---

**Algorithm 1** SAM-SGA for ZSAQ
___
**Input:** generator $\mathcal{G}_{\boldsymbol{\theta}}$, quantized model $\mathcal{Q}(\hat{\boldsymbol{\omega}})$, pretrained model $\mathcal{P}$, learning rate $\eta_{\boldsymbol{\theta}}$ and $\eta_{\boldsymbol{\omega}}$, and the neighborhood size $\beta$ for the perturbation
1: **Initialize:** $\mathcal{Q}(\hat{\boldsymbol{\omega}}_0)$ is quantized from $\mathcal{P}$ by Eq.(1)
2: **for** t=0,1,...,T **do**
3:     //training quantized model by SAM
4:     Compute the optimal perturbation $\epsilon^*(\hat{\boldsymbol{\omega}})$ by Eq.(5);
5:     Calculate the gradient and update $\boldsymbol{\omega}_{t+1}$ via Eq.(7);
6:     Quantize $\boldsymbol{\omega}_{t+1}$ to $\hat{\boldsymbol{\omega}}_{t+1}$ by Eq .(1);
7:     //training generator by SGA
8:     Update $\boldsymbol{\theta}_{t+1}$ by Eq.(6);
9: **end for**
**Output:** $\hat{\boldsymbol{\omega}}$ uniformly drawn from $\{\hat{\boldsymbol{\omega}}_1, ..., \hat{\boldsymbol{\omega}}_T\}$.
___

$\mathbb{E}_{\mathbf{z} \sim \mathcal{T}} \mathcal{D}(\mathcal{P}(\mathcal{G}_{\boldsymbol{\theta}}(\mathbf{z})), \mathcal{Q}(\hat{\boldsymbol{\omega}}; \mathcal{G}_{\boldsymbol{\theta}}(\mathbf{z})))$ with a general representation $f(\hat{\boldsymbol{\omega}}, \boldsymbol{\theta}) = \mathbb{E}_{\xi}[f(\hat{\boldsymbol{\omega}}, \boldsymbol{\theta}; \xi)]$, where $\xi$ follows i.i.d. Then our detailed objective function (3) can be turned into:

$$\min_{\hat{\boldsymbol{\omega}}} \{ \max_{\boldsymbol{\theta}} f(\hat{\boldsymbol{\omega}}, \boldsymbol{\theta}) + f^{sharp}(\hat{\boldsymbol{\omega}}, \boldsymbol{\theta}) \}, \quad (8)$$

where $f^{sharp}(\hat{\boldsymbol{\omega}}, \boldsymbol{\theta}) \triangleq f^{SAM}(\hat{\boldsymbol{\omega}}, \boldsymbol{\theta}) - f(\hat{\boldsymbol{\omega}}, \boldsymbol{\theta})$ and $f^{SAM}(\hat{\boldsymbol{\omega}}, \boldsymbol{\theta}) \triangleq \max_{\|\epsilon\|_2 \leq \beta} f(\hat{\boldsymbol{\omega}} + \epsilon, \boldsymbol{\theta})$. We can solve the maximization problem that $\epsilon^*(\hat{\boldsymbol{\omega}}) = \beta \frac{\nabla_{\hat{\boldsymbol{\omega}}} f(\hat{\boldsymbol{\omega}}, \boldsymbol{\theta})}{\|\nabla_{\hat{\boldsymbol{\omega}}} f(\hat{\boldsymbol{\omega}}, \boldsymbol{\theta})\|_2}$. So that we can write $f^{SAM}(\hat{\boldsymbol{\omega}}, \boldsymbol{\theta})$ as $f(\hat{\boldsymbol{\omega}} + \epsilon^*(\hat{\boldsymbol{\omega}}), \boldsymbol{\theta})$.

Next, we restate the update rule of our Algorithm 1 in our new denotation that:

$$\begin{cases} \boldsymbol{\omega}_{t+1} = \hat{\boldsymbol{\omega}}_t - \eta_{\boldsymbol{\omega}} g_{\boldsymbol{\omega}}(\hat{\boldsymbol{\omega}}_t + \beta \frac{g_{\boldsymbol{\omega}}(\hat{\boldsymbol{\omega}}_t, \boldsymbol{\theta}_t)}{\|g_{\boldsymbol{\omega}}(\hat{\boldsymbol{\omega}}_t, \boldsymbol{\theta}_t)\|_2}, \boldsymbol{\theta}_t) \\ \hat{\boldsymbol{\omega}}_{t+1} = Quantization(\boldsymbol{\omega}_{t+1}) \\ \boldsymbol{\theta}_{t+1} = \boldsymbol{\theta}_t + \eta_{\boldsymbol{\theta}} g_{\boldsymbol{\theta}}(\hat{\boldsymbol{\omega}}_t, \boldsymbol{\theta}_t) \end{cases},$$

where $g_{\boldsymbol{\omega}}$ and $g_{\boldsymbol{\theta}}$ are the approximated gradients calculated by $g_{\boldsymbol{\omega}}(\boldsymbol{\omega}, \boldsymbol{\theta}) = \frac{1}{M} \sum_{i=1}^{M} G_{\boldsymbol{\omega}}(\boldsymbol{\omega}, \boldsymbol{\theta}; \xi_i)$ and $g_{\boldsymbol{\theta}}(\boldsymbol{\omega}, \boldsymbol{\theta}) = \frac{1}{M} \sum_{i=1}^{M} G_{\boldsymbol{\theta}}(\boldsymbol{\omega}, \boldsymbol{\theta}; \xi_i)$ respectively. And $G \triangleq (G_{\boldsymbol{\omega}}, G_{\boldsymbol{\theta}})$ is the unbiasd gradient of the function $f$, i.e., $\mathbb{E}[G(\boldsymbol{\omega}, \boldsymbol{\theta}; \xi)] \in \nabla f(\boldsymbol{\omega}, \boldsymbol{\theta})$.
**Remark.** Actually function $f$ maps $\mathbb{R}^d \times \mathbb{R}^n$ to $\mathbb{R}$. The first parameter can be either a floating point or an integer. It is well-defined for us to represent $f(\hat{\boldsymbol{\omega}}, \cdot)$ or $\nabla_{\boldsymbol{\omega}} f(\hat{\boldsymbol{\omega}}, \cdot)$.

Before illustrating the theoretical results, we first introduce some necessary assumptions and definitions that are used in the analysis.

**Assumption 4.1.** $f(\boldsymbol{\omega}, \boldsymbol{\theta})$ *is differential and $l$-Lipschitz smooth:*

$$\|\nabla_{\boldsymbol{\omega}} f(\boldsymbol{\omega}_1, \boldsymbol{\theta}) - \nabla_{\boldsymbol{\omega}} f(\boldsymbol{\omega}_2, \boldsymbol{\theta})\| \le l\|\boldsymbol{\omega}_1 - \boldsymbol{\omega}_2\|, \quad \forall \boldsymbol{\theta}$$
$$\|\nabla_{\boldsymbol{\theta}} f(\boldsymbol{\omega}, \boldsymbol{\theta}_1) - \nabla_{\boldsymbol{\theta}} f(\boldsymbol{\omega}, \boldsymbol{\theta}_2)\| \le l\|\boldsymbol{\theta}_1 - \boldsymbol{\theta}_2\|, \quad \forall \boldsymbol{\omega}$$

**Assumption 4.2.** $f(\boldsymbol{\omega}, \cdot)$ *satisfies PL condition on every given $\boldsymbol{\omega}$, i.e. there exists $\mu > 0$ such that $\|\nabla_{\boldsymbol{\theta}} f(\boldsymbol{\omega}, \boldsymbol{\theta})\|^2 \ge 2\mu[\max_{\boldsymbol{\theta}} f(\boldsymbol{\omega}, \boldsymbol{\theta}) - f(\boldsymbol{\omega}, \boldsymbol{\theta})]$.*

**Assumption 4.3.** *Parameter $\boldsymbol{\theta}$ is restricted in a convex and bounded set with a diameter of $D$.*

**Assumption 4.4.** *The approximated gradient is unbiased and has a bounded variance, i.e.,*

$$\mathbb{E}[G(\boldsymbol{\omega}, \boldsymbol{\theta}, \xi)] \in \nabla f(\boldsymbol{\omega}, \boldsymbol{\theta});$$
$$\mathbb{E}\|G(\boldsymbol{\omega}, \boldsymbol{\theta}, \xi) - \nabla f(\boldsymbol{\omega}, \boldsymbol{\theta})\|^2 \le \sigma^2.$$

**Remark.** These assumptions are common in the context of minimax optimization problems and can be realized in empirical practice.

**Lemma 4.1.** *(Zhu et al., 2020) The quantization error is bounded by the grain, where $d$ denotes the dimension of parameter $\boldsymbol{\omega}$:*

$$\|\hat{\boldsymbol{\omega}} - \boldsymbol{\omega}\| \le \sqrt{d}\Delta.$$

**Remark.** The quantization error can be naturally derived from Eq. (1) and the grain $\Delta$ is decided by default quantization setting, which will not be discussed in our paper.

**Definition 4.1.** $\Phi(\boldsymbol{\omega}) = \max_{\boldsymbol{\theta}} f(\boldsymbol{\omega}, \boldsymbol{\theta})$; $\Phi^* = \min_{\boldsymbol{\omega}} \Phi(\boldsymbol{\omega})$. *And we define an $\epsilon$-stationary point when $\mathbb{E}\|\nabla \Phi(\boldsymbol{\omega})\|^2 \le \epsilon$.*

**Remark.** By defining the function $\Phi$, we can simplify our minimax optimization with two opposite variables into a minimization optimization problem with a single variable. When the gradient of $\Phi(\boldsymbol{\omega})$ diminishes to zero, we consider our algorithm converges.

**Theorem 4.1.** *Under Assumption 4.1,4.2,4.3,4.4 and restrictions $\beta \le \frac{\eta_{\boldsymbol{\theta}}}{2l}$, $\eta_{\boldsymbol{\theta}} = 64\kappa^2\eta_{\boldsymbol{\omega}}$, $\eta_{\boldsymbol{\omega}} \le \min\{\frac{1}{128\kappa^2 l}, \sqrt{\frac{M(\mathbb{E}[\Phi(\hat{\boldsymbol{\omega}}_0)] - \mathbb{E}[\Phi^*])}{132T\kappa^4 l\sigma^2}}\}$, we have the convergence bound for our problem:*

$$\frac{1}{T}\sum_{t=0}^{T-1} \mathbb{E}\|\nabla\Phi(\hat{\boldsymbol{\omega}})\|^2 \tag{9}$$
$$\le 2\sqrt{\frac{(\Phi(\hat{\boldsymbol{\omega}}_0) - \Phi^*)\kappa^4\sigma^2}{MT}} + \mathcal{O}(d\Delta^2).$$

**Remark.** Due to the space limitation, the proofs and strongly-concave case are placed in Appendix A. Observing the above results, we evaluate the averaged gradients at the quantized parameter $\hat{\boldsymbol{\omega}}$ and we can draw the conclusion that our averaged output of the quantized models can converge to an $\epsilon$-stationary point during $\mathcal{O}(1/\epsilon^2)$ iterations under the neglect of quantization error $d\Delta^2$.

## 5 Experiment

### 5.1 Setup

**Tasks and Models.** In this section, we evaluate our SAM-SGA method on both discriminative (*i.e.*, BERT-style) and generative (*i.e.*, GPT-style) language models. For discriminative models, we followed many prior works (Zhong et al., 2023c,d) and tested both BERT$_{\text{base}}$ and BERT$_{\text{large}}$ (Devlin et al., 2018) on GLUE benchmark (Wang et al., 2018); and for generative models, we tested the OPT-family (Zhang et al., 2022) (mainly OPT-350m) on the GLUE benchmark and several language modeling tasks, *i.e.*, WikiText2 (Merity et al.) and Penn Treebank (PTB) (Mikolov and Zweig, 2012) and WikiText103 (Merity et al.). For evaluation, we report the performance with Accuracy ("*Acc.*") metric for most tasks, except the Pearson correlation ("*Pear.*") for STS-B, the Matthew correlation ("*Mcc.*") for CoLA, the perplexity (PPL) score for language modeling tasks. We report the averaged results over 5 random seeds to avoid stochasticity. The details of all tasks are shown in Appendix A.1.

**Implementation Details.** Following many previous studies (Tao et al., 2022; Bai et al., 2021b), we first fine-tune a full-precision model using the pre-trained checkpoint from huggingface[1] for each task, and use the fine-tuned model as the full-precision teacher model. Then, we apply the cutting-edge PTQ methods, *i.e.*, ZeroQuant (Yao et al., 2022) for BERT models and GPTQ (Frantar et al., 2022) for OPT models, to quantize the fine-tuned model and use the quantized network to initialize the student model. For the generator model, we directly use the OPT-350m (Zhang et al., 2022) (without any further tuning) model[2], i.e., the generators are not trained on any end-task data. All experiments are conducted on a single NVIDIA A100 (40G) GPU

---

[1] https://huggingface.co/models

[2] Notably, we can use any generative LLMs as the generator models. In this work, we use the OPT models as they are open-sourced and cover multiple model scales.

| Method | #Bits (W-A) | CoLA Mcc. | MNLI Acc. | MRPC Acc. | QNLI Acc. | QQP Pear. | RTE Acc. | SST2 Acc. | STSB Acc. | GLUE Avg. | Δ (↑) |
|---|---|---|---|---|---|---|---|---|---|---|---|
| Full-precision | W32A32 | 60.82 | 83.12 | 85.05 | 90.52 | 89.91 | 65.34 | 92.43 | 88.84 | 82.00 | - |
| Baseline | W8A8 | 60.07 | 83.16 | 84.31 | 90.41 | 89.86 | 64.98 | 92.66 | 88.82 | 81.78 | * |
| QAT-GT | W8A8 | 62.26 | **83.58** | **85.29** | **90.57** | 89.87 | **66.79** | 92.66 | 88.85 | **82.48** | +0.70 |
| QAT-Rand | W8A8 | 60.06 | 83.37 | 84.81 | 90.52 | 89.92 | 65.14 | **93.00** | **88.98** | 81.98 | +0.20 |
| **SAM-SGA** | W8A8 | **63.70** | 82.93 | 85.05 | 90.52 | **89.93** | 65.34 | 92.08 | 88.96 | 82.31 | **+0.81** |
| Baseline | W4A8 | 52.13 | 77.24 | 82.60 | 85.10 | 86.34 | 50.90 | 89.11 | 84.41 | 75.98 | * |
| QAT-GT | W4A8 | **57.30** | **78.22** | 83.09 | **86.95** | **86.41** | 55.86 | **90.71** | **87.34** | 78.24 | +2.26 |
| QAT-Rand | W4A8 | 53.37 | 76.98 | 77.94 | 86.61 | 86.23 | 48.38 | 90.48 | 86.48 | 75.81 | -0.17 |
| **SAM-SGA** | W4A8 | 53.01 | 76.64 | **83.58** | 86.12 | 86.16 | **64.26** | 89.68 | 86.94 | **78.30** | **+2.32** |
| Baseline | W2A8 | 0.00 | 35.47 | 68.38 | 49.13 | 67.29 | 47.29 | 77.52 | 58.43 | 50.44 | * |
| QAT-GT | W2A8 | 0.00 | **39.86** | **69.61** | 50.59 | 67.55 | 47.29 | **77.64** | 69.41 | 52.74 | +2.30 |
| QAT-Rand | W2A8 | 0.00 | 39.26 | 31.62 | 49.52 | 67.93 | 47.29 | 76.83 | 63.11 | 46.95 | -3.49 |
| **SAM-SGA** | W2A8 | **10.16** | 36.84 | 68.38 | **53.41** | **89.95** | **57.76** | 76.26 | 66.61 | **57.42** | **+6.98** |

Table 1: Results of BERT_base on the development set of GLUE (Wang et al., 2018) benchmark. "#Bits (W-A)" denotes the bit-width for weights of Transformer layers and activations.

| Method | #Bits (W-A) | CoLA Mcc. (↑) | MRPC Acc. (↑) | RTE Acc. (↑) |
|---|---|---|---|---|
| Full-precision | W32A32 | 64.47 | 86.03 | 65.71 |
| Baseline | W8A8 | 64.80 | 86.76 | 62.45 |
| QAT-GT | W8A8 | 66.61 | **87.75** | 65.34 |
| QAT-Rand | W8A8 | 66.86 | 86.52 | 65.15 |
| **SAM-SGA** | W8A8 | 66.08 | 87.25 | **66.06** |
| Baseline | W4A8 | 49.11 | 74.26 | 47.29 |
| QAT-GT | W4A8 | 50.85 | 75.73 | **56.68** |
| QAT-Rand | W4A8 | 45.63 | 70.34 | 47.29 |
| **SAM-SGA** | W4A8 | **52.97** | **77.70** | 52.71 |

Table 2: Results of BERT_large on parts of tasks of GLUE (Wang et al., 2018) benchmark.

| Method | #Bits (W-A) | PTB PPL (↓) | WikiText103 PPL (↓) | WikiText2 PPL (↓) |
|---|---|---|---|---|
| Full-precision | W32A32 | 28.58 | 13.38 | 19.21 |
| Baseline | W8A8 | 56.02 | 48.16 | 55.4 |
| QAT-GT | W8A8 | 50.68 | **42.15** | **46.29** |
| QAT-Rand | W8A8 | 54.59 | 48.69 | 60.46 |
| **SAM-SGA** | W8A8 | **46.59** | 46.56 | 54.08 |
| Baseline | W4A4 | 60.25 | 56.16 | 62.89 |
| QAT-GT | W4A4 | 55.59 | **42.01** | **51.72** |
| QAT-Rand | W4A4 | 59.40 | 55.32 | 63.63 |
| **SAM-SGA** | W4A4 | **51.19** | 54.32 | 61.22 |

Table 3: Results of OPT-350m on the dev set of four language modeling benchmarks.

and the detailed hyper-parameters can be found in Appendix A.2.

**Compared Methods.** For reference, we compare our SAM-SGA method with the following cutting-edge counterparts:

- **Full-precision**: we report the results of full-precision fine-tuned model, *i.e.*, teacher model in our framework.

- **Baseline**: we adopt the powerful PTQ methods *i.e.*, ZeroQuant (Yao et al., 2022) for BERT models and GPTQ (Frantar et al., 2022) for OPT models, to quantize the fine-tuned model, and report the results as the baseline.

- **QAT-GT**: after obtaining the teacher and student models, we use the original training data to guide the quantization-aware training process of student model.

- **QAT-Rand**: we follow Yao et al. (2022) and use the random data (using the random integer number to generate token ids) to perform the QAT process.

Notably, for each method, we carefully grid-search the best learning rates from {5e-6, 1e-5, 2e-5, 5e-5}, and report the best results that are chosen from the best single run among those learning rates.

## 5.2 Main Results

We report the results of BERT_base, BERT_large and OPT-350m in Table 1, 2 and 3, respectively. Notably, we use WxAy to represent the x-bit weight quantization and y-bit activation quantization. From the results, we can observe that:

**SAM-SGA outperforms the other counterparts in most settings.** As shown in Table 1, our SAM-SGA outperforms the baseline methods among all quantization settings by a large margin, *i.e.*,

| Method | CoLA | MNLI | MRPC | QNLI | QQP | RTE | SST2 | STSB | GLUE | |
| | Mcc. | Acc. | Acc. | Acc. | Pear. | Acc. | Acc. | Acc. | Avg. | Δ (↑) |
| --- | --- | --- | --- | --- | --- | --- | --- | --- | --- | --- |
| Baseline | 52.13 | 77.24 | 82.60 | 85.10 | 86.34 | 50.90 | 89.11 | 84.41 | 75.98 | * |
| **SAM-SGA (Ours)** | 53.01 | 76.64 | 83.58 | 86.12 | 86.16 | 64.26 | 89.68 | 86.94 | 78.30 | **+2.32** |
| -w/o SGA | 45.30 | 78.06 | 84.31 | 85.53 | 86.05 | 55.23 | 89.36 | 86.59 | 76.30 | -2.00 |
| -w/o SAM | 50.74 | 76.29 | 82.60 | 85.74 | 85.93 | 63.53 | 89.44 | 86.46 | 77.59 | -0.71 |
| -w/o SGA&SAM | 43.70 | 76.02 | 82.80 | 85.33 | 85.56 | 56.04 | 89.48 | 86.11 | 75.63 | -2.67 |

Table 4: Ablation study of important components of SAM-SGA. Here, we use the BERT-base models quantified into W4A8 bit-width.

up to +6.98 average score. Specifically, QAT-GT also achieves remarkable performance gains, owing to the supervised information from the original training data. When using the random data ("QAT-Rand"), the performance gains brought by the vanilla QAT process will be much slighter. Encouragingly, it can be found that ZSAQ performs better in the lower precision settings, indicating the potential of validation methods in extreme compression scenarios. These results prove the effectiveness and significance of our data-free method.

**SAM-SGA brings the consistent performance gains among both model sizes.** We further adopt our SAM-SGA to the larger BERT model. Due to the space limitation, we only report the results of parts of GLUE benchmarks in Table 2, and provide the full results in Table 7 of Appendix A.3. It can be found the similar phenomenon in the BERT-base experiments. That is, our SAM-SGA brings the consistent performance gains on both models.

**SAM-SGA works well in both BERT-style and GPT-style models.** In addition to the discriminative PLMs, we also verify the effectiveness of our SAM-SGA on the generative models. The results of language modeling tasks are in Table 3, and we can seen that, with the help of SAM-SGA, OPT-350m achieves much better quantization performance against the baselines. Moreover, we also evaluate the OPT-350m on the GLUE benchmark. The contrastive results are listed in Table 8 of Appendix A.3, illustrating that our method outperforms the other methods, which further validates the effectiveness of our methods for generative style models.

## 5.3 Ablation Study

In this part, we 1) first evaluate the impact of important components of our SAM-SGA, and 2) then investigate the effect of different generator models.

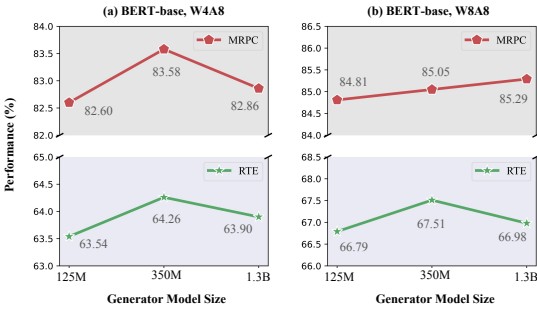

Figure 2: Ablation study of different generator models. We use OPT-family models with different sizes as the generators.

**Impact of important components of SAM-SGA.** There are several important components in our SAM-SGA framework, *i.e.*, "SGA": tuning the generator model with the stochastic gradient ascent, "SAM": training the quantized model with SAM optimizer. Here, we conduct the ablation study to verify whether these components are necessary. Specifically, taking the BERT-base model as an example, we compare our full SAM-SGA method with the following methods: i) "-w/o SGA": we remove the SGA process, *i.e.*, keeping the generator model fixed; ii) "-w/o SAM": we replace the SAM optimizer with the vanilla AdamW optimizer; 3) "-w/o SGA&SAM": fixed generator model and base optimizer are used.

As shown in Table 4, we can find that removing any component will cause the performance degradation, indicating the necessarily of these components. More specifically, without the SGA, our SAM-SGA would perform much worse. One possible reason is that the quantized model might over-fit the synthetic data, without the constraint of adversarial training.

**Effect of different generator models.** Here, we examine whether our SAM-SGA can still work well using different generator models. We con-

| Method | #Bits | GLUE | |
| --- | --- | --- | --- |
| | (W-A) | Avg. | Δ (↑) |
| Full-precision | W32A32 | 82.00 | * |
| PTQ-vanilla | W8A8 | 79.81 | -2.19 |
| PTQ-PEG | W8A8 | 81.06 | -0.94 |
| PTQ-Adaround | W8A8 | 82.02 | +0.02 |
| **SAM-SGA (Ours)** | W8A8 | **82.59** | **+0.59** |
| PTQ-vanilla | W4A8 | 35.07 | -46.93 |
| PTQ-PEG | W4A8 | 39.46 | -42.54 |
| PTQ-Adaround | W4A8 | 77.49 | -4.51 |
| **SAM-SGA (Ours)** | W4A8 | **78.30** | **-3.70** |

Table 5: Comparisons with other zero-shot quantization methods on the development set of GLUE (Wang et al., 2018) benchmark. Full results are shown in Table 9.

| Method | #Bits | PTB | WikiText2 |
| --- | --- | --- | --- |
| | (W-A) | PPL (↓) | PPL (↓) |
| Full-precison | W32A32 | 24.88 | 15.52 |
| Baseline | W8A8 | 36.75 | 24.05 |
| SAM-SGA | W8A8 | 32.66 | 20.96 |
| Δ (↓) | - | ↓ **4.09** | ↓ **3.09** |
| Baseline | W4A4 | 44.46 | 27.54 |
| SAM-SGA | W4A4 | 36.62 | 22.88 |
| Δ (↓) | - | ↓ **7.84** | ↓ **4.66** |

Table 6: Results of OPT-1.3b on two language modeling benchmarks. We can find that our SAM-SGA can still work well in the billion-level model scenarios.

duct the contrastive experiments by varying the generator model size from 120M to 1.3B, and illustrate the results in Figure 2. As seen, compared to the baseline, our SAM-SGA consistently brings improvements across all model sizes, basically indicating that the performance of SAM-SGA is not very sensitive to the generator model size. More specifically, the case of OPT-350m performs best, and we thereby use this setting in our experiment.

### 5.4 Analysis and Discussion

Here, we conduct extensive analyses to discuss: 1) whether our SAM-SGA outperforms the other zero-shot quantization counterparts; 2) whether our SAM-SGA can expand to more larger language model scenarios, and 3) whether it gains better model generalization.

**Comparisons with other zero-shot quantization methods.** To better assess the strengths of our method, we further compare it with more zero-shot quantization methods, i.e., PTQ-vanilla (vanilla post-training quantization), PTQ-PEG (Bondarenko et al., 2021) and PTQ-

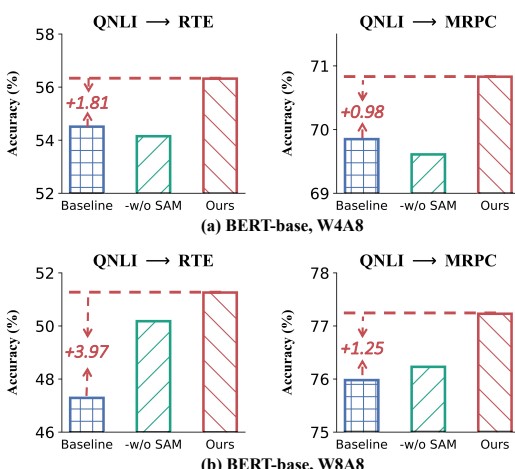

Figure 3: Analysis of task generalization. The model is fine-tuned on the QNLI task and transferred to four different tasks. We can see that our method consistently brings better generalization compared to the baseline.

Adaround (Nagel et al., 2020). Specifically, taking the BERT-base as an example, we show the contrastive results in the W8A8 and W4A8 settings in Table 5. As seen, our method SAM-SGA outperforms the other zero-shot quantization methods by a clear margin, especially in the low-bit setting.

**Scaling to Billion-level large models.** Some readers may concern that whether our method can be used to quantify the larger language models. Here, we attempt to expand our method to the quantization of billion-level models. Specifically, due to the limited amount of computing resources, we have tried our best to extend SAM-SGA to a larger model setting, and lastly choose the OPT-1.3b for experiments. For reference, we compared our method with the baseline GPTQ method[3].

As shown in Table 6, compared to the baseline, our SAM-SGA brings the significant and consistent performance improvements among all settings, i.e., up to +7.84 gains. These results prove that our method also works well in the larger language model scenarios.

**Does SAM-SGA Bring Better Generalization?** One of our main contributions is that we introduce the SAM optimizer to improve the generalization of quantized model. Here, we verify it from two perspectives: i) measuring the cross-task zero-shot

---

[3]Notably, we empirically found that the performance of GPTQ is very sensitive to the number of input samples (default is 1), especially in the large model scenarios. Here, we set this number to 10 to ensure the effectiveness of GPTQ.

performance, and *ii*) visualizing the loss landscapes of quantization models.

**Task Generalization.** As stated in many previous studies (Zhong et al., 2023b; Ding et al., 2022), the performance of out-of-domain (OOD) data is widely used to verify the model generalization. Hence, we follow Xu et al. (2021); Zhong et al. (2022a) and evaluate the performance of quantized model on several OOD data. In practice, we first quantize BERT-base models (fine-tuned on QNLI task) with different methods, including "Baseline", "Ours" and its variant "-w/o SAM" (without using SAM). Then, we inference the quantified models on other tasks, *i.e.*, MRPC and RTE. The results are illustrated in Figure 3. We observe that "Ours" consistently outperforms the other counterparts. To be more specific, compared with baseline, our SAM-SGA brings a +2.0 average improvement score on all tasks, indicating that ***our SAM-SGA boosts the performance of PLMs on OOD data.***

**Visualization of Landscape.** To have a close look, we visualize the loss landscapes of different quantized OPT-350m models fine-tuned on the PTB task. In practice, we follow He et al. (2021); Zhong et al. (2022a) to plot the 1D loss curve by linear interpolation between the model weights before (denoted as $\theta_0$) and after (denoted as $\theta_1$) tuning, *i.e.*, "$\theta_1 + \alpha \cdot (\theta_1 - \theta_0)$", where $\alpha$ is a scalar parameter that is ranged from -1 to 1. The 1D visualization results are illustrated in Figure 4, and we find that our optimal setting "Ours" shows a flatter and optimal property. ***These results prove that our SAM-SGA can smooth the loss landscape and improve the generalization of PLMs effectively.***

☞ **A Note on More Analyses and Discussions**

Notably, in addition to the above results and studies, we further conduct more in-depth and systematic analyses and discussions in Appendix, due to space limitations. Specifically, we provide 1) parameter analysis of $\eta$, and 2) visualization of the loss curve of adversarial training stages in Appendix A.3. Please refer to the Appendix for more details.

## 6 Conclusion

In this paper, we propose a novel zero-shot quantization framework ZSAQ, which effectively realizes zero-shot quantization without much accuracy drop and shows promising generalization ability. We

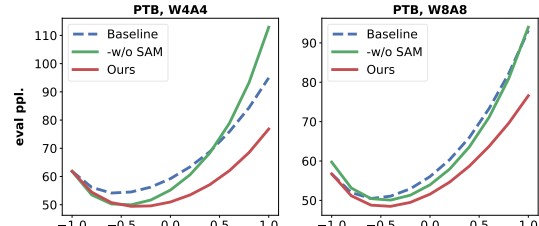

Figure 4: 1D visualization of loss landscapes of OPT-350m quantized by different methods. PTB (Mikolov and Zweig, 2012) is used for evaluation.

provide theoretical analysis for the minimax optimization algorithm under both nonconvex-strongly-concave and nonconvex-PL conditions, which guarantees a convergence rate of $\mathcal{O}(1/\sqrt{T})$ for our ZSAQ method. Our experiments show outperforming quantization results, up to +6.98 average score, and validate improved generalization ability.

## Limitations

Our work has several potential limitations. First, given the limited computational budget, we only validate our SAM-SGA method on the Large and Base model sizes. It will make our work more convincing if scaling the experiments up to the much larger model sizes, *e.g.*, LLaMA-65b (Touvron et al., 2023) and OPT-66b. On the other hand, although our method brings much performance gains compared to the other PTQ method, it would lead to more computation overheads. How to accelerate the process has not been explored in this work.

## Ethics Statement

We take ethical considerations very seriously, and strictly adhere to the EMNLP Ethics Policy. This paper focuses on zero-shot quantization for current open-sourced pretrained language models, but not capturing the privacy knowledge. Both the models and evaluation datasets used in this paper are publicly available and have been widely adopted by researchers. Therefore, we believe that this research will not pose ethical issues.

## Acknowledgements

This work was supported in part by the National Natural Science Foundation of China under Grants 62225113 and 62076186. The numerical calculations in this paper have been done on the supercomputing system in the Supercomputing Center of Wuhan University.

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

# A Appendix

## A.1 Details of Tasks and Datasets

In this work, we conduct extensive experiments on GLUE benchmark and some widely-used language modeling tasks. Here, we introduce the descriptions of the used tasks and datasets in detail:

**CoLA** Corpus of Linguistic Acceptability (Warstadt et al., 2019) is a binary single-sentence classification task to determine whether a given sentence is linguistically "acceptable".

**MRPC** Microsoft Research Paraphrase Corpus (Dolan and Brockett, 2005) is a task to predict whether two sentences are semantically equivalent.

**STS-B** Semantic Textual Similarity (Cer et al., 2017) is a task to predict how similar two sentences are on a 1-5 scale in terms of semantic meaning.

**RTE** Recognizing Textual Entailment (Giampiccolo et al., 2007), given a premise and a hypothesis, is a task to predict whether the premise entails the hypothesis.

**MNLI** The Multi-Genre Natural Language Inference Corpus (Williams et al., 2018) is a task to predict whether the premise entails the hypothesis, contradicts the hypothesis, or neither, given a premise sentence and a hypothesis sentence.

**SST-2** The Stanford Sentiment Treebank (Socher et al., 2013) is a binary classification task to predict the sentiment of a given sentence.

**QNLI** Question Natural Language Inference is a binary classification task constructed from SQuAD (Rajpurkar et al., 2016), which aims to predict whether a context sentence contains the answer to a question sentence.

**QQP** The Quora Question Pairs2 dataset is a collection of question pairs. The task is to determine whether a pair of questions are semantically equivalent.

**PTB** The English Penn Treebank (PTB) (Mikolov and Zweig, 2012) corpus is one of the most known and used corpus for the evaluation of models for sequence labelling, and is also commonly used for character-level and word-level Language Modelling.

**WikiText2 and WikiText103** Compared to the preprocessed version of Penn Treebank (PTB), WikiText-2 (Merity et al.) is over 2 times larger and WikiText-103 (Merity et al.) is over 110 times larger. The WikiText datasets are also widely-used for language modeling. As they are composed of full articles, the datasets are well suited for models that can take advantage of long term dependencies.

## A.2 Details of Hyper-parameters

BERT and OPT models are obtained from Huggingface[4], and we follow the instruction (*e.g.*, fine-tuneing hyper-parameters) from Huggingface Transformer Library[5] to fine-tune these models.

For initial post-training quantization of BERT-based models, we use the ZeroQuant (Yao et al., 2022) and set 48 groups for group-wise weight quantization in W8A8 settings, 32 groups for W4A8 and 16 groups for W2A8. While for OPT-based models, we use 128 groups in all settings for GPTQ (Frantar et al., 2022) method.

For our SAM-SGA method, we use 50 iterations with batch size 32 and sequence length 128 for BERT models, and we use 100 iterations with batch size 4 and sequence length 2048 for OPT models. We grid search the learning rate in range of {1e-6, 5r-6, 1e-5, 2e-5}. All the quantized models are trained using a single NVIDIA A100 (40G) GPU.

## A.3 More Results and Analyses

In this part, we provide more results and analyses to further investigate the effectiveness of our method. Specifically, we first perform the parameter analyses on $\eta_\theta$ and $\eta_\omega$, and then visualize the loss curve of adversarial training stages.

**Parameter Analyses of $\eta_\theta$ and $\eta_\omega$.** The factors $\eta_\theta$ and $\eta_\omega$ are two important hyper-parameters in our Algorithm. In this study, we analyze its influence by evaluating the performance of BERT-base (W4A8) with different $\eta$ spanning {5e-6,1e-5,2e-5,5e-5} on MNLI and QNLI tasks. Figure 5 illustrates the grid-search results. We find that the results do not exhibit significant fluctuations while demonstrating stability within a narrow range. These prove the effectiveness of our method.

**Visualization of loss curve of adversarial training stages.** Since adversarial training usually suffers from instability, some readers may concern about the training stability of our method. To investigate it, we visualize the loss curve of adversarial training stages in Figure 6. We can see the student_loss decreasing and generator_loss increasing as time varies and eventually reaching a relatively stable plateau during the final stage. These results indicate that the adversarial training is relatively stable in our SAM-SGA framework.

---

[4] https://huggingface.co/models
[5] https://github.com/huggingface/transformers/tree/main/examples/pytorch

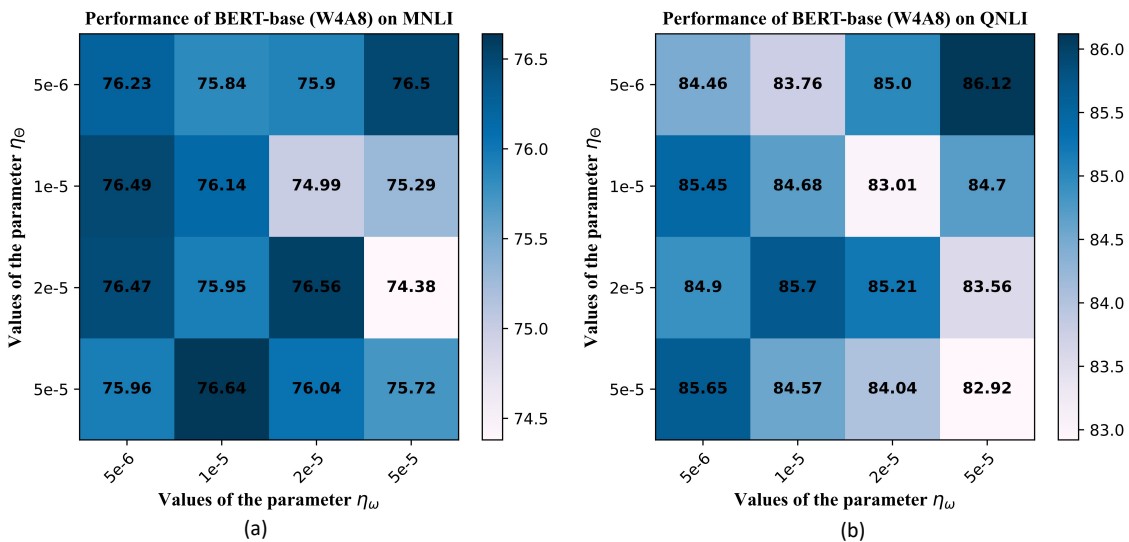

Figure 5: Parameter analyses of $\eta_\theta$ and $\eta_\omega$. We report the performance of BERT-base (W4A8) on MNLI and QNLI.

| Method | #Bits | CoLA | MNLI | MRPC | QNLI | QQP | RTE | SST2 | STSB | GLUE | |
|---|---|---|---|---|---|---|---|---|---|---|---|
| | (W-A) | Mcc. | Acc. | Acc. | Acc. | Pear. | Acc. | Acc. | Acc. | Avg. | Δ (↑) |
| Full-precision | W32A32 | 64.47 | 83.12 | 86.03 | 92.37 | 90.85 | 65.71 | 93.35 | 89.22 | 83.14 | - |
| Baseline | W8A8 | 64.80 | 83.33 | 86.76 | 92.27 | 90.79 | 62.45 | 93.46 | 89.12 | 82.87 | - |
| QAT-GT | W8A8 | **66.61** | 83.29 | **87.75** | 92.54 | 90.81 | 65.34 | 93.23 | 89.26 | 83.60 | +0.73 |
| QAT-Rand | W8A8 | 66.86 | 83.01 | 86.52 | 92.42 | 90.76 | 65.15 | 93.00 | 89.18 | 83.36 | +0.49 |
| **SAM-SGA** | W8A8 | 66.08 | **83.50** | 87.25 | **92.55** | **90.87** | **66.06** | **94.04** | **89.37** | **83.72** | **+0.85** |
| Baseline | W4A8 | 49.11 | 78.86 | 74.26 | 86.86 | 86.68 | 47.29 | 87.73 | 87.63 | 74.80 | - |
| QAT-GT | W4A8 | 50.85 | 79.17 | 75.73 | 90.44 | **87.35** | **56.68** | **92.32** | 88.05 | **77.57** | +2.77 |
| QAT-Rand | W4A8 | 45.63 | 79.02 | 70.34 | 87.26 | 87.17 | 47.29 | 78.84 | **88.08** | 72.95 | -1.85 |
| **SAM-SGA** | W4A8 | **52.97** | **79.19** | **77.70** | **90.69** | 87.20 | 52.71 | 90.48 | 87.72 | 77.33 | +2.53 |

Table 7: Full comparison results of BERT$_{\text{large}}$ on the development set of GLUE (Wang et al., 2018) benchmark.

## A.4 Convergence Analysis

Before our proof, we give some notations for brief. $\widetilde{\omega}_{t+1/2} \triangleq \hat{\omega}_t + \beta g_\omega(\hat{\omega}_t, \theta_t)$ like in (Foret et al., 2020). So the update rule for $\omega$ can be rewritten as: $\omega_{t+1} = \hat{\omega}_t - \eta_\omega g_\omega(\widetilde{\omega}_{t+1/2}, \theta_t)$.

**Lemma A.1.** *Under Assumption 4.4, we can further get the evaluation about the approximate function:*

$$\mathbb{E}[g(\omega, \theta)] \in \nabla f(\omega, \theta)$$

$$\mathbb{E}\|g(\omega, \theta) - \nabla f(\omega, \theta)\|^2 \leq \frac{\sigma^2}{M}$$

*Proof.* According to the definition that $g(\omega, \theta) = \frac{1}{M}\sum_{i=1}^{M} G(\omega, \theta, \xi_i)$, and $\xi_i$ is i.i.d. So the expectation of the approximate function $g$ is the same as $G$ that: $\mathbb{E}[g(\omega, \theta)] \in \nabla f(\omega, \theta)$ and

$$\mathbb{E}\|g(\omega, \theta) - \nabla f(\omega, \theta)\|^2$$

$$= \mathbb{E}\|\frac{1}{M}\sum_{i=1}^{M} G(\omega, \theta, \xi_i) - \nabla f(\omega, \theta)\|^2$$

$$= \frac{1}{M^2}\sum_{i=1}^{M} \mathbb{E}\|G(\omega, \theta, \xi_i) - \nabla f(\omega, \theta)\|^2 \leq \frac{\sigma^2}{M}$$

$\square$

**Lemma A.2.** *We give an estimation that*

$$\mathbb{E}\|g_\omega(\widetilde{\omega}_{t+1/2}, \theta_t)\|^2$$

$$\leq (4\beta^2 l^2 + 2\beta l + 2)\mathbb{E}\|\nabla_\omega f(\hat{\omega}_t, \theta_t)\|^2 + (5\beta^2 l^2 + 2)\frac{\sigma^2}{M}$$

*Proof.* We have the following decomposition:

$$\mathbb{E}\|g_\omega(\widetilde{\omega}_{t+1/2}, \theta_t)\|^2$$
$$= \mathbb{E}\|g_\omega(\widetilde{\omega}_{t+1/2}, \theta_t) - \nabla_\omega f(\hat{\omega}_t, \theta_t)\|^2$$
$$- \mathbb{E}\|\nabla_\omega f(\hat{\omega}_t, \theta_t)\|^2$$
$$+ 2\mathbb{E}\langle g_\omega(\widetilde{\omega}_{t+1/2}, \theta_t), \nabla_\omega f(\hat{\omega}_t, \theta_t)\rangle \tag{10}$$

| Method | #Bits (W-A) | CoLA Mcc. | MNLI Acc. | MRPC Acc. | QNLI Acc. | QQP Pear. | RTE Acc. | SST2 Acc. | STSB Acc. | GLUE Avg. | GLUE Δ (↑) |
|---|---|---|---|---|---|---|---|---|---|---|---|
| Full-precision | W32A32 | 57.51 | 84.54 | 83.33 | 90.30 | 90.72 | 69.68 | 93.23 | 88.13 | 82.18 | - |
| Baseline | W4A8 | 55.69 | 84.03 | 82.60 | 89.73 | 90.25 | 67.15 | 92.77 | 87.93 | 81.27 | * |
| QAT-GT | W4A8 | 55.94 | 83.85 | **83.58** | 89.85 | **90.32** | **68.23** | 93.12 | 88.03 | 81.62 | +0.35 |
| QAT-Rand | W4A8 | 55.71 | 83.75 | 82.60 | 89.77 | 90.17 | 67.51 | 92.78 | 88.03 | 81.29 | +0.02 |
| **SAM-SGA** | W4A8 | **56.47** | **83.94** | 83.33 | **89.96** | 90.27 | 67.87 | **93.23** | 88.03 | **81.64** | +0.37 |

Table 8: Results of OPT-350m on the development set of GLUE (Wang et al., 2018) benchmark. Here, we only report the results in W4A8 settings, and note that the results in the other low-bit settings are similar.

| Method | #Bits (W-A) | CoLA Mcc. | MNLI Acc. | MRPC Acc. | QNLI Acc. | QQP Pear. | RTE Acc. | SST2 Acc. | STSB Acc. | GLUE Avg. |
|---|---|---|---|---|---|---|---|---|---|---|
| Full-precision | W32A32 | 60.82 | 83.12 | 85.05 | 90.52 | 89.91 | 65.34 | 92.43 | 88.84 | 82.00 |
| PTQ-vanilla | W8A8 | 58.29 | 82.08 | 80.64 | 88.72 | 87.47 | 62.82 | 90.37 | 88.10 | 79.81 |
| PTQ-PEG | W8A8 | 57.04 | 83.15 | **85.29** | 89.95 | 89.36 | 64.62 | 90.71 | 88.34 | 81.06 |
| PTQ-Adaround | W8A8 | 61.32 | **83.68** | 84.07 | 90.43 | 89.84 | 66.06 | **92.32** | 88.45 | 82.02 |
| **SAM-SGA** | W8A8 | **63.70** | 82.93 | 85.05 | **90.52** | **89.93** | **67.51** | 92.08 | **88.96** | **82.59** |
| PTQ-vanilla | W4A8 | 1.88 | 33.91 | 31.62 | 50.06 | 62.97 | 52.71 | 48.62 | -1.23 | 35.07 |
| PTQ-PEG | W4A8 | 0.00 | 37.91 | 31.62 | 50.23 | 63.18 | 47.29 | 73.62 | 11.83 | 39.46 |
| PTQ-Adaround | W4A8 | 55.67 | 72.59 | 79.90 | **87.13** | **86.49** | **64.62** | **91.28** | 82.21 | 77.49 |
| **SAM-SGA** | W4A8 | **53.01** | **76.64** | **83.58** | 86.12 | 86.16 | 64.26 | 89.68 | **86.94** | **78.30** |

Table 9: Results of comparisons with other zero-shot quantization methods on the development set of GLUE (Wang et al., 2018) benchmark.

For the cross-product term, we evaluate it as follows:

$$
\begin{aligned}
&\mathbb{E}\langle g_{\boldsymbol{\omega}}(\widetilde{\boldsymbol{\omega}}_{t+1/2}, \boldsymbol{\theta}_t), \nabla_{\boldsymbol{\omega}} f(\hat{\boldsymbol{\omega}}_t, \boldsymbol{\theta}_t)\rangle \\
&= \mathbb{E}\langle g_{\boldsymbol{\omega}}(\widetilde{\boldsymbol{\omega}}_{t+1/2}, \boldsymbol{\theta}_t) \\
&\quad - g_{\boldsymbol{\omega}}(\hat{\boldsymbol{\omega}}_t + \beta\nabla_{\boldsymbol{\omega}} f(\hat{\boldsymbol{\omega}}_t, \boldsymbol{\theta}_t), \boldsymbol{\theta}_t), \nabla_{\boldsymbol{\omega}} f(\hat{\boldsymbol{\omega}}_t, \boldsymbol{\theta}_t)\rangle \\
&\quad + \mathbb{E}\langle g_{\boldsymbol{\omega}}(\hat{\boldsymbol{\omega}}_t + \beta\nabla_{\boldsymbol{\omega}} f(\hat{\boldsymbol{\omega}}_t, \boldsymbol{\theta}_t), \boldsymbol{\theta}_t), \nabla_{\boldsymbol{\omega}} f(\hat{\boldsymbol{\omega}}_t, \boldsymbol{\theta}_t)\rangle \\
&= \mathbb{E}\langle \nabla_{\boldsymbol{\omega}} f(\hat{\boldsymbol{\omega}}_t + \beta\nabla_{\boldsymbol{\omega}} f(\hat{\boldsymbol{\omega}}_t, \boldsymbol{\theta}_t), \boldsymbol{\theta}_t), \nabla_{\boldsymbol{\omega}} f(\hat{\boldsymbol{\omega}}_t, \boldsymbol{\theta}_t)\rangle \\
&\quad + \mathbb{E}\langle \nabla_{\boldsymbol{\omega}} f(\widetilde{\boldsymbol{\omega}}_{t+1/2}, \boldsymbol{\theta}_t) - \nabla_{\boldsymbol{\omega}} f(\hat{\boldsymbol{\omega}}_t + \beta\nabla_{\boldsymbol{\omega}} f(\hat{\boldsymbol{\omega}}_t, \boldsymbol{\theta}_t), \boldsymbol{\theta}_t), \\
&\qquad \nabla_{\boldsymbol{\omega}} f(\hat{\boldsymbol{\omega}}_t, \boldsymbol{\theta}_t)\rangle \\
&\leq \frac{1}{2}\mathbb{E}\|\nabla_{\boldsymbol{\omega}} f(\widetilde{\boldsymbol{\omega}}_{t+1/2}, \boldsymbol{\theta}_t) - \nabla_{\boldsymbol{\omega}} f(\hat{\boldsymbol{\omega}}_t + \beta\nabla_{\boldsymbol{\omega}} f(\hat{\boldsymbol{\omega}}_t, \boldsymbol{\theta}_t), \boldsymbol{\theta}_t)\|^2 \\
&\quad + \frac{1}{2}\mathbb{E}\|\nabla_{\boldsymbol{\omega}} f(\hat{\boldsymbol{\omega}}_t, \boldsymbol{\theta}_t)\|^2 + \mathbb{E}\|\nabla_{\boldsymbol{\omega}} f(\hat{\boldsymbol{\omega}}_t, \boldsymbol{\theta}_t)\|^2 \\
&\quad + \mathbb{E}\langle \nabla_{\boldsymbol{\omega}} f(\hat{\boldsymbol{\omega}}_t + \beta\nabla_{\boldsymbol{\omega}} f(\hat{\boldsymbol{\omega}}_t, \boldsymbol{\theta}_t), \boldsymbol{\theta}_t) - \nabla_{\boldsymbol{\omega}} f(\hat{\boldsymbol{\omega}}_t, \boldsymbol{\theta}_t), \\
&\qquad \nabla_{\boldsymbol{\omega}} f(\hat{\boldsymbol{\omega}}_t, \boldsymbol{\theta}_t)\rangle \\
&\leq (\beta l + \frac{3}{2})\mathbb{E}\|\nabla_{\boldsymbol{\omega}} f(\hat{\boldsymbol{\omega}}_t, \boldsymbol{\theta}_t)\|^2 + \frac{\beta^2 l^2 \sigma^2}{2M}
\end{aligned}
\tag{11}
$$

And for the first term, we have:

$$
\begin{aligned}
&\mathbb{E}\|g_{\boldsymbol{\omega}}(\widetilde{\boldsymbol{\omega}}_{t+1/2}, \boldsymbol{\theta}_t) - \nabla_{\boldsymbol{\omega}} f(\hat{\boldsymbol{\omega}}_t, \boldsymbol{\theta}_t)\|^2 \\
&\leq 2\mathbb{E}\|g_{\boldsymbol{\omega}}(\widetilde{\boldsymbol{\omega}}_{t+1/2}, \boldsymbol{\theta}_t) - \nabla_{\boldsymbol{\omega}} f(\widetilde{\boldsymbol{\omega}}_{t+1/2}, \boldsymbol{\theta}_t)\|^2 \\
&\quad + 2\mathbb{E}\|\nabla_{\boldsymbol{\omega}} f(\widetilde{\boldsymbol{\omega}}_{t+1/2}, \boldsymbol{\theta}_t) - \nabla_{\boldsymbol{\omega}} f(\hat{\boldsymbol{\omega}}_t, \boldsymbol{\theta}_t)\|^2 \\
&\leq \frac{2\sigma^2}{M} + 2l^2 \mathbb{E}\|\widetilde{\boldsymbol{\omega}}_{t+1/2} - \hat{\boldsymbol{\omega}}_t\|^2 \\
&\leq 2\frac{\sigma^2}{M}(2\beta^2 l^2 + 1) + 4\beta^2 l^2 \mathbb{E}\|\nabla_{\boldsymbol{\omega}} f(\hat{\boldsymbol{\omega}}_t, \boldsymbol{\theta}_t)\|^2
\end{aligned}
\tag{12}
$$

Combining the above inequalities and we can get:

$$
\begin{aligned}
&\mathbb{E}\|g_{\boldsymbol{\omega}}(\widetilde{\boldsymbol{\omega}}_{t+1/2}, \boldsymbol{\theta}_t)\|^2 \\
&\leq (4\beta^2 l^2 + 2\beta l + 2)\mathbb{E}\|\nabla_{\boldsymbol{\omega}} f(\hat{\boldsymbol{\omega}}_t, \boldsymbol{\theta}_t)\|^2 \\
&\quad + (5\beta^2 l^2 + 2)\frac{\sigma^2}{M}
\end{aligned}
\tag{13}
$$

$\square$

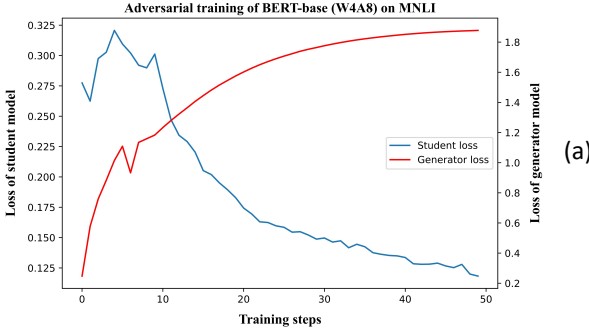

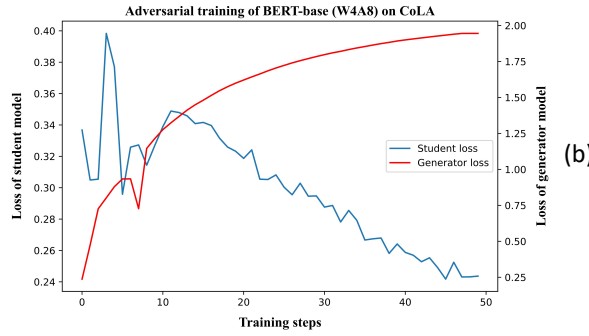

Figure 6: Visualization of loss curve of adversarial training stages.

### A.5 Strongly Concave Case

**Lemma A.3.** *For the descending relationship of the function $\Phi$, we have:*

$$\mathbb{E}[\Phi(\hat{\boldsymbol{\omega}}_{t+1})]$$
$$\leq \mathbb{E}[\Phi(\hat{\boldsymbol{\omega}}_t)]$$
$$-\frac{\eta_{\boldsymbol{\omega}}}{2}(\frac{15}{16}-5\beta l-4\kappa l\eta_{\boldsymbol{\omega}}(4\beta^2 l^2+2\beta l+2))\mathbb{E}\|\nabla\Phi(\hat{\boldsymbol{\omega}}_t)\|^2$$
$$+[\frac{\eta_{\boldsymbol{\omega}}}{2}(1+\frac{1}{2}\beta l)+2\kappa l\eta_{\boldsymbol{\omega}}^2(4\beta^2 l^2+2\beta l+2)]\mathbb{E}[\|\nabla\Phi(\hat{\boldsymbol{\omega}}_t)-$$
$$\nabla_{\boldsymbol{\omega}}f(\hat{\boldsymbol{\omega}}_t,\boldsymbol{\theta}_t)\|^2]+(5\beta^2 l^2+2)\frac{\kappa l\eta_{\boldsymbol{\omega}}^2\sigma^2}{M}$$
$$+\frac{\beta l\sigma^2\eta_{\boldsymbol{\omega}}}{M}+(2\kappa l+\frac{8}{\eta_{\boldsymbol{\omega}}})d\Delta^2$$

*Proof.* We get the conclusion that $\Phi(\boldsymbol{\omega})$ is $2\kappa l$-smooth according to Lemma 4.3 in (Lin et al., 2020):

$$\Phi(\hat{\boldsymbol{\omega}}_{t+1})$$
$$\leq \Phi(\hat{\boldsymbol{\omega}}_t)+\langle\nabla\Phi(\hat{\boldsymbol{\omega}}_t),\hat{\boldsymbol{\omega}}_{t+1}-\hat{\boldsymbol{\omega}}_t\rangle+\kappa l\|\hat{\boldsymbol{\omega}}_{t+1}-\hat{\boldsymbol{\omega}}_t\|^2$$
$$\leq \Phi(\hat{\boldsymbol{\omega}}_t)+\langle\nabla\Phi(\hat{\boldsymbol{\omega}}_t),\boldsymbol{\omega}_{t+1}-\hat{\boldsymbol{\omega}}_t\rangle+2\kappa l\|\boldsymbol{\omega}_{t+1}-\hat{\boldsymbol{\omega}}_t\|^2$$
$$+\langle\nabla\Phi(\hat{\boldsymbol{\omega}}_t),\hat{\boldsymbol{\omega}}_{t+1}-\boldsymbol{\omega}_{t+1}\rangle+2\kappa l d\Delta^2$$
$$=\Phi(\hat{\boldsymbol{\omega}}_t)-\eta_{\boldsymbol{\omega}}\langle\nabla\Phi(\hat{\boldsymbol{\omega}}_t),g_{\boldsymbol{\omega}}(\widetilde{\boldsymbol{\omega}}_{t+1/2},\boldsymbol{\theta}_t)\rangle+2\kappa l d\Delta^2$$
$$+2\kappa l\eta_{\boldsymbol{\omega}}^2\|g_{\boldsymbol{\omega}}(\widetilde{\boldsymbol{\omega}}_{t+1/2},\boldsymbol{\theta}_t)\|^2$$
$$+\frac{\eta_{\boldsymbol{\omega}}}{32}\|\nabla\Phi(\boldsymbol{\omega}_t)\|^2+\frac{8}{\eta_{\boldsymbol{\omega}}}d\Delta^2$$
$$\tag{14}$$

Taking expectation conditioned on $(\boldsymbol{\omega}_t,\boldsymbol{\theta}_t)$ and we get:

$$\mathbb{E}[\Phi(\hat{\boldsymbol{\omega}}_{t+1})|\boldsymbol{\omega}_t,\boldsymbol{\theta}_t]$$
$$\leq \Phi(\hat{\boldsymbol{\omega}}_t)-\eta_{\boldsymbol{\omega}}\langle\nabla\Phi(\hat{\boldsymbol{\omega}}_t),\nabla_{\boldsymbol{\omega}}f(\widetilde{\boldsymbol{\omega}}_{t+1/2},\boldsymbol{\theta}_t)\rangle$$
$$+2\kappa l\eta_{\boldsymbol{\omega}}^2\mathbb{E}[\|g_{\boldsymbol{\omega}}(\widetilde{\boldsymbol{\omega}}_{t+1/2},\boldsymbol{\theta}_t)\|^2|\boldsymbol{\omega}_t,\boldsymbol{\theta}_t]$$
$$+\frac{\eta_{\boldsymbol{\omega}}}{32}\mathbb{E}\|\nabla\Phi(\hat{\boldsymbol{\omega}}_t)\|^2+(2\kappa l+\frac{8}{\eta_{\boldsymbol{\omega}}})d\Delta^2$$
$$\tag{15}$$

We again take expectations on both sides of the above inequality so we have:

$$\mathbb{E}[\Phi(\hat{\boldsymbol{\omega}}_{t+1})]$$
$$\leq \mathbb{E}[\Phi(\hat{\boldsymbol{\omega}}_t)]-\eta_{\boldsymbol{\omega}}\mathbb{E}\langle\nabla\Phi(\hat{\boldsymbol{\omega}}_t),\nabla_{\boldsymbol{\omega}}f(\widetilde{\boldsymbol{\omega}}_{t+1/2},\boldsymbol{\theta}_t)\rangle$$
$$+2\kappa l\eta_{\boldsymbol{\omega}}^2\mathbb{E}\|g_{\boldsymbol{\omega}}(\widetilde{\boldsymbol{\omega}}_{t+1/2},\boldsymbol{\theta}_t)\|^2+\frac{\eta_{\boldsymbol{\omega}}}{32}\mathbb{E}\|\nabla\Phi(\hat{\boldsymbol{\omega}}_t)\|^2$$
$$+(2\kappa l+\frac{8}{\eta_{\boldsymbol{\omega}}})d\Delta^2$$
$$\tag{16}$$

For the second term, we decompose it as follows:

$$\mathbb{E}\langle\nabla\Phi(\hat{\boldsymbol{\omega}}_t),\nabla_{\boldsymbol{\omega}}f(\widetilde{\boldsymbol{\omega}}_{t+1/2},\boldsymbol{\theta}_t)\rangle$$
$$=\mathbb{E}\langle\nabla\Phi(\hat{\boldsymbol{\omega}}_t),$$
$$\nabla_{\boldsymbol{\omega}}f(\hat{\boldsymbol{\omega}}_t,\boldsymbol{\theta}_t)+\nabla_{\boldsymbol{\omega}}f(\widetilde{\boldsymbol{\omega}}_{t+1/2},\boldsymbol{\theta}_t)-\nabla_{\boldsymbol{\omega}}f(\hat{\boldsymbol{\omega}}_t,\boldsymbol{\theta}_t)\rangle$$
$$\geq \mathbb{E}\langle\nabla\Phi(\hat{\boldsymbol{\omega}}_t),\nabla_{\boldsymbol{\omega}}f(\hat{\boldsymbol{\omega}}_t,\boldsymbol{\theta}_t)\rangle$$
$$-\mathbb{E}\|\nabla\Phi(\hat{\boldsymbol{\omega}}_t)\|\|\nabla_{\boldsymbol{\omega}}f(\widetilde{\boldsymbol{\omega}}_{t+1/2},\boldsymbol{\theta}_t)-\nabla_{\boldsymbol{\omega}}f(\hat{\boldsymbol{\omega}}_t,\boldsymbol{\theta}_t)\|$$
$$\geq\mathbb{E}\langle\nabla\Phi(\hat{\boldsymbol{\omega}}_t),\nabla_{\boldsymbol{\omega}}f(\hat{\boldsymbol{\omega}}_t,\boldsymbol{\theta}_t)\rangle-\beta l\mathbb{E}\|\nabla\Phi(\hat{\boldsymbol{\omega}}_t)\|\|g_{\boldsymbol{\omega}}(\hat{\boldsymbol{\omega}}_t,\boldsymbol{\theta}_y)\|$$
$$\geq \mathbb{E}\langle\nabla\Phi(\hat{\boldsymbol{\omega}}_t),\nabla\Phi(\hat{\boldsymbol{\omega}}_t)+\nabla_{\boldsymbol{\omega}}f(\hat{\boldsymbol{\omega}}_t,\boldsymbol{\theta}_t)-\nabla\Phi(\hat{\boldsymbol{\omega}}_t)\rangle$$
$$-\beta l\mathbb{E}\|\nabla\Phi(\hat{\boldsymbol{\omega}}_t)\|(\|\nabla_{\boldsymbol{\omega}}f(\hat{\boldsymbol{\omega}}_t,\boldsymbol{\theta}_t)\|+\|g_{\boldsymbol{\omega}}(\hat{\boldsymbol{\omega}}_t,\boldsymbol{\theta}_t)-\nabla_{\boldsymbol{\omega}}f(\hat{\boldsymbol{\omega}}_t,\boldsymbol{\theta}_t)\|)$$
$$\geq \mathbb{E}\|\nabla\Phi(\hat{\boldsymbol{\omega}}_t)\|^2-\frac{1}{2}\mathbb{E}\|\nabla\Phi(\hat{\boldsymbol{\omega}}_t)\|^2$$
$$-\frac{1}{2}\mathbb{E}\|\nabla_{\boldsymbol{\omega}}f(\hat{\boldsymbol{\omega}}_t,\boldsymbol{\theta}_t)-\nabla\Phi(\hat{\boldsymbol{\omega}}_t)\|^2$$
$$-\beta l\mathbb{E}\|\nabla\Phi(\hat{\boldsymbol{\omega}}_t)\|\|\nabla_{\boldsymbol{\omega}}f(\hat{\boldsymbol{\omega}}_t,\boldsymbol{\theta}_t)\|-\frac{1}{2}\beta l\mathbb{E}\|\nabla\Phi(\hat{\boldsymbol{\omega}}_t)\|^2$$
$$-\frac{1}{2}\beta l\mathbb{E}\|g_{\boldsymbol{\omega}}(\hat{\boldsymbol{\omega}}_t,\boldsymbol{\theta}_t)-\nabla_{\boldsymbol{\omega}}f(\hat{\boldsymbol{\omega}}_t,\boldsymbol{\theta}_t)\|^2$$
$$\geq\frac{1-\beta l}{2}\mathbb{E}\|\nabla\Phi(\hat{\boldsymbol{\omega}}_t)\|^2-\frac{1}{2}\mathbb{E}\|\nabla_{\boldsymbol{\omega}}f(\hat{\boldsymbol{\omega}}_t,\boldsymbol{\theta}_t)-\nabla\Phi(\hat{\boldsymbol{\omega}}_t)\|^2$$
$$-\beta l\mathbb{E}\|\nabla\Phi(\hat{\boldsymbol{\omega}}_t)\|\|\nabla_{\boldsymbol{\omega}}f(\hat{\boldsymbol{\omega}}_t,\boldsymbol{\theta}_t)\|-\frac{\beta l\sigma^2}{2M}$$
$$\tag{17}$$

We continue estimating the last term in above inequality (17):

$$\mathbb{E}\|\nabla\Phi(\hat{\boldsymbol{\omega}}_t)\|\|\nabla_{\boldsymbol{\omega}}f(\hat{\boldsymbol{\omega}}_t,\boldsymbol{\theta}_t)\|$$
$$=\mathbb{E}\|\nabla\Phi(\hat{\boldsymbol{\omega}}_t)\|\|\nabla_{\boldsymbol{\omega}}f(\hat{\boldsymbol{\omega}}_t,\boldsymbol{\theta}_t)-\nabla\Phi(\hat{\boldsymbol{\omega}}_t)+\nabla\Phi(\hat{\boldsymbol{\omega}}_t)\|$$
$$\leq\mathbb{E}\|\nabla\Phi(\hat{\boldsymbol{\omega}}_t)\|^2+\mathbb{E}\|\nabla\Phi(\hat{\boldsymbol{\omega}}_t)\|\|\nabla_{\boldsymbol{\omega}}f(\hat{\boldsymbol{\omega}}_t,\boldsymbol{\theta}_t)-\nabla\Phi(\hat{\boldsymbol{\omega}}_t)\|$$
$$\overset{(i)}{\leq}\mathbb{E}\|\nabla\Phi(\hat{\boldsymbol{\omega}}_t)\|^2+\mathbb{E}\|\nabla\Phi(\hat{\boldsymbol{\omega}}_t)\|^2+\frac{1}{4}\mathbb{E}\|\nabla_{\boldsymbol{\omega}}f(\hat{\boldsymbol{\omega}}_t,\boldsymbol{\theta}_t)-\nabla\Phi(\hat{\boldsymbol{\omega}}_t)\|^2$$
$$\tag{18}$$

where the last inequality $(i)$ is due to Young's inequality.

Combining inequality (17) with (18), we can get:

$$\mathbb{E}\langle\nabla\Phi(\hat{\boldsymbol{\omega}}_t),\nabla_{\boldsymbol{\omega}}f(\widetilde{\boldsymbol{\omega}}_{t+1/2},\boldsymbol{\theta}_t)\rangle$$

$$\geq\frac{1-\beta l}{2}\mathbb{E}\|\nabla\Phi(\hat{\boldsymbol{\omega}}_t)\|^2-\frac{1}{2}\mathbb{E}\|\nabla_{\boldsymbol{\omega}}f(\hat{\boldsymbol{\omega}}_t,\boldsymbol{\theta}_t)-\nabla\Phi(\hat{\boldsymbol{\omega}}_t)\|^2$$

$$-2\beta l\mathbb{E}\|\nabla\Phi(\hat{\boldsymbol{\omega}}_t)\|^2-\frac{\beta l}{4}\mathbb{E}\|\nabla_{\boldsymbol{\omega}}f(\hat{\boldsymbol{\omega}}_t,\boldsymbol{\theta}_t)-\nabla\Phi(\hat{\boldsymbol{\omega}}_t)\|^2-\frac{\beta l\sigma^2}{2M}$$

$$=\frac{1}{2}(1-5\beta l)\mathbb{E}\|\nabla\Phi(\hat{\boldsymbol{\omega}}_t)\|^2-\frac{1}{2}(1+\frac{1}{2}\beta l)\mathbb{E}[\|\nabla\Phi(\hat{\boldsymbol{\omega}}_t)-$$

$$\nabla_{\boldsymbol{\omega}}f(\hat{\boldsymbol{\omega}}_t,\boldsymbol{\theta}_t)\|^2]-\frac{\beta l\sigma^2}{2M} \tag{19}$$

Finally, we combine inequality (16) with Lemma A.2 and inequality (19):

$$\mathbb{E}[\Phi(\hat{\boldsymbol{\omega}}_{t+1})]$$

$$\leq\mathbb{E}[\Phi(\hat{\boldsymbol{\omega}}_t)]-\frac{\eta_{\boldsymbol{\omega}}}{2}(1-5\beta l)\mathbb{E}\|\nabla\Phi(\hat{\boldsymbol{\omega}}_t)\|^2$$

$$+\frac{\eta_{\boldsymbol{\omega}}}{2}(1+\frac{1}{2}\beta l)\mathbb{E}\|\nabla\Phi(\hat{\boldsymbol{\omega}}_t)-\nabla_{\boldsymbol{\omega}}f(\hat{\boldsymbol{\omega}}_t,\boldsymbol{\theta}_t)\|^2$$

$$+2\kappa l\eta_{\boldsymbol{\omega}}^2((4\beta^2l^2+2\beta l+2)\mathbb{E}\|\nabla_{\boldsymbol{\omega}}f(\hat{\boldsymbol{\omega}}_t,\boldsymbol{\theta}_t)\|^2$$

$$+(5\beta^2l^2+2)\frac{\sigma^2}{M})+\frac{\beta l\sigma^2\eta_{\boldsymbol{\omega}}}{M}$$

$$+\frac{\eta_{\boldsymbol{\omega}}}{32}\mathbb{E}\|\nabla\Phi(\hat{\boldsymbol{\omega}}_t)\|^2+(2\kappa l+\frac{8}{\eta_{\boldsymbol{\omega}}})d\Delta^2$$

$$\overset{(i)}{\leq}\mathbb{E}[\Phi(\hat{\boldsymbol{\omega}}_t)]$$

$$-\frac{\eta_{\boldsymbol{\omega}}}{2}(\frac{15}{16}-5\beta l-4\kappa l\eta_{\boldsymbol{\omega}}(4\beta^2l^2+2\beta l+2))\mathbb{E}\|\nabla\Phi(\hat{\boldsymbol{\omega}}_t)\|^2$$

$$+[\frac{\eta_{\boldsymbol{\omega}}}{2}(1+\frac{1}{2}\beta l)+2\kappa l\eta_{\boldsymbol{\omega}}^2(4\beta^2l^2+2\beta l+2)]\mathbb{E}[\|\nabla\Phi(\hat{\boldsymbol{\omega}}_t)-$$

$$\nabla_{\boldsymbol{\omega}}f(\hat{\boldsymbol{\omega}}_t,\boldsymbol{\theta}_t)\|^2]+(5\beta^2l^2+2)\frac{\kappa l\eta_{\boldsymbol{\omega}}^2\sigma^2}{M}$$

$$+\frac{\beta l\sigma^2\eta_{\boldsymbol{\omega}}}{M}+(2\kappa l+\frac{8}{\eta_{\boldsymbol{\omega}}})d\Delta^2 \tag{20}$$

where the last inequality $(i)$ uses the Cauchy-Schwarz inequality that $\|\nabla_{\boldsymbol{\omega}}f(\hat{\boldsymbol{\omega}}_t,\boldsymbol{\theta}_t)\|^2\leq 2\|\nabla\Phi(\hat{\boldsymbol{\omega}}_t)\|^2+2\|\nabla_{\boldsymbol{\omega}}f(\hat{\boldsymbol{\omega}}_t,\boldsymbol{\theta}_t)-\nabla\Phi(\hat{\boldsymbol{\omega}}_t)\|^2$. $\square$

**Lemma A.4.** *When we require the learning rate satisfy: $\eta_{\boldsymbol{\omega}}(2\beta l+1)\leq\frac{1}{16\kappa^2l}$ and $\eta_{\boldsymbol{\theta}}=\frac{1}{2l}$, we have the evaluation about the error term:*

$$\mathbb{E}\|\boldsymbol{\theta}^*(\hat{\boldsymbol{\omega}}_{t+1})-\boldsymbol{\theta}_{t+1}\|^2$$

$$\leq\lambda^{t+1}D^2$$

$$+4(4\kappa-1)\kappa^2\eta_{\boldsymbol{\omega}}^2(4\beta^2l^2+2\beta l+2)\sum_{j=0}^{t}\lambda^{t-j}\mathbb{E}\|\nabla\Phi(\hat{\boldsymbol{\omega}}_j)\|^2$$

$$+((4\kappa-1)\kappa^2\eta_{\boldsymbol{\omega}}^2(5\beta^2l^2+2)+\frac{4\kappa-1}{4\kappa-2}\eta_{\boldsymbol{\theta}}^2)\frac{\sigma^2}{M}\sum_{j=0}^{t}\lambda^j$$

$$+2(4\kappa-1)\kappa^2d\Delta^2\sum_{j=0}^{t}\lambda^j$$

*where $\lambda=1-\frac{1}{4\kappa}+4(4\kappa-1)\kappa^2l^2\eta_{\boldsymbol{\omega}}^2(4\beta^2l^2+2\beta l+2)$.*
*Proof.*

$$\mathbb{E}\|\boldsymbol{\theta}^*(\hat{\boldsymbol{\omega}}_{t+1})-\boldsymbol{\theta}_{t+1}\|^2$$

$$=\mathbb{E}\|\boldsymbol{\theta}^*(\hat{\boldsymbol{\omega}}_{t+1})-\boldsymbol{\theta}^*(\hat{\boldsymbol{\omega}}_t)+\boldsymbol{\theta}_{t+1}-\boldsymbol{\theta}^*(\hat{\boldsymbol{\omega}}_t)\|^2$$

$$\leq(4\kappa-1)\mathbb{E}\|\boldsymbol{\theta}^*(\hat{\boldsymbol{\omega}}_{t+1})-\boldsymbol{\theta}^*(\hat{\boldsymbol{\omega}}_t)\|^2$$

$$+(1+\frac{1}{4\kappa-2})\mathbb{E}\|\boldsymbol{\theta}_{t+1}-\boldsymbol{\theta}^*(\hat{\boldsymbol{\omega}}_t)\|^2 \tag{21}$$

where the inequality is due to Young's inequality.

We then continue to decompose the second term as follows according to the update rule and take expectaion:

$$\mathbb{E}\|\boldsymbol{\theta}_{t+1}-\boldsymbol{\theta}^*(\hat{\boldsymbol{\omega}}_t)\|^2$$

$$=\mathbb{E}\|\mathcal{P}_{\Theta}(\boldsymbol{\theta}_t+\eta_{\boldsymbol{\theta}}g_{\boldsymbol{\theta}}(\hat{\boldsymbol{\omega}}_t,\boldsymbol{\theta}_t))-\boldsymbol{\theta}^*(\hat{\boldsymbol{\omega}}_t)\|^2$$

$$\leq\mathbb{E}\|\boldsymbol{\theta}_t-\boldsymbol{\theta}^*(\hat{\boldsymbol{\omega}}_t)\|^2+\eta_{\boldsymbol{\theta}}^2\mathbb{E}\|g_{\boldsymbol{\theta}}(\hat{\boldsymbol{\omega}}_t,\boldsymbol{\theta}_t)\|^2$$

$$+2\eta_{\boldsymbol{\theta}}\mathbb{E}\langle\boldsymbol{\theta}_t-\boldsymbol{\theta}^*(\hat{\boldsymbol{\omega}}_t),g_{\boldsymbol{\theta}}(\hat{\boldsymbol{\omega}}_t,\boldsymbol{\theta}_t)\rangle$$

$$\leq\mathbb{E}\|\boldsymbol{\theta}_t-\boldsymbol{\theta}^*(\hat{\boldsymbol{\omega}}_t)\|^2+\eta_{\boldsymbol{\theta}}^2\mathbb{E}\|\nabla_{\boldsymbol{\theta}}f(\hat{\boldsymbol{\omega}}_t,\boldsymbol{\theta}_t)\|^2$$

$$+2\eta_{\boldsymbol{\theta}}\mathbb{E}\langle\boldsymbol{\theta}_t-\boldsymbol{\theta}^*(\hat{\boldsymbol{\omega}}_t),\nabla_{\boldsymbol{\theta}}f(\hat{\boldsymbol{\omega}}_t,\boldsymbol{\theta}_t)\rangle+\frac{\eta_{\boldsymbol{\theta}}^2\sigma^2}{M} \tag{22}$$

Using the property of $\mu$-strong concavity of function $f$ on variable $\boldsymbol{\theta}$, we can get:

$$f(\hat{\boldsymbol{\omega}}_t,\boldsymbol{\theta}_t)-f(\hat{\boldsymbol{\omega}}_t,\boldsymbol{\theta}^*(\hat{\boldsymbol{\omega}}_t))-\langle\boldsymbol{\theta}_t-\boldsymbol{\theta}^*(\hat{\boldsymbol{\omega}}_t),\nabla_{\boldsymbol{\theta}}f(\hat{\boldsymbol{\omega}}_t,\boldsymbol{\theta}_t)\rangle$$

$$\geq\frac{\mu}{2}\|\boldsymbol{\theta}^*(\hat{\boldsymbol{\omega}}_t)-\boldsymbol{\theta}_t\|^2 \tag{23}$$

So the original inequality (22) above can be turned into:

$$\mathbb{E}\|\boldsymbol{\theta}_{t+1}-\boldsymbol{\theta}^*(\hat{\boldsymbol{\omega}}_t)\|^2$$

$$\leq(1-\mu\eta_{\boldsymbol{\theta}})\mathbb{E}\|\boldsymbol{\theta}_t-\boldsymbol{\theta}^*(\hat{\boldsymbol{\omega}}_t)\|^2+\eta_{\boldsymbol{\theta}}^2\mathbb{E}\|\nabla_{\boldsymbol{\theta}}f(\hat{\boldsymbol{\omega}}_t,\boldsymbol{\theta}_t)\|^2$$

$$+2\eta_{\boldsymbol{\theta}}\mathbb{E}[f(\hat{\boldsymbol{\omega}}_t,\boldsymbol{\theta}_t)-f(\hat{\boldsymbol{\omega}}_t,\boldsymbol{\theta}^*(\hat{\boldsymbol{\omega}}_t))]+\frac{\eta_{\boldsymbol{\theta}}^2\sigma^2}{M} \tag{24}$$

Since $f(\boldsymbol{\omega},\cdot)$ is $l$-smooth and $\mu$-strongly concave, we have:

$$f(\hat{\boldsymbol{\omega}}_t,\boldsymbol{\theta}_t)-f(\hat{\boldsymbol{\omega}}_t,\boldsymbol{\theta}^*(\hat{\boldsymbol{\omega}}_t))+\frac{1}{2l}\|\nabla_{\boldsymbol{\theta}}f(\hat{\boldsymbol{\omega}}_t,\boldsymbol{\theta}_t)-\nabla_{\boldsymbol{\theta}}f(\hat{\boldsymbol{\omega}}_t,\boldsymbol{\theta}^*(\hat{\boldsymbol{\omega}}_t))\|^2$$

$$\leq\langle\nabla_{\boldsymbol{\theta}}f(\hat{\boldsymbol{\omega}}_t,\boldsymbol{\theta}^*(\hat{\boldsymbol{\omega}}_t)),\boldsymbol{\theta}_t-\boldsymbol{\theta}^*(\hat{\boldsymbol{\omega}}_t)\rangle \tag{25}$$

On the other hand, we can get the following inequality according to optimal condition:

$$\langle\nabla_{\boldsymbol{\theta}}f(\hat{\boldsymbol{\omega}}_t,\boldsymbol{\theta}^*(\hat{\boldsymbol{\omega}}_t)),\boldsymbol{\theta}-\boldsymbol{\theta}^*(\hat{\boldsymbol{\omega}}_t)\rangle\leq 0;\quad\forall\boldsymbol{\theta}\in\Theta \tag{26}$$

Combining above two inequalities (25)(26), we have:

$$f(\hat{\boldsymbol{\omega}}_t, \boldsymbol{\theta}_t) - f(\hat{\boldsymbol{\omega}}_t, \boldsymbol{\theta}^*(\hat{\boldsymbol{\omega}}_t))$$
$$\leq -\frac{1}{2l}\|\nabla_{\boldsymbol{\theta}} f(\hat{\boldsymbol{\omega}}_t, \boldsymbol{\theta}_t) - \nabla_{\boldsymbol{\theta}} f(\hat{\boldsymbol{\omega}}_t, \boldsymbol{\theta}^*(\hat{\boldsymbol{\omega}}_t))\|^2$$
(27)

Putting pieces (24) and (27) together and on the occasion that $\eta_{\boldsymbol{\theta}} = \frac{1}{2l}$ we can get:

$$\mathbb{E}\|\boldsymbol{\theta}_{t+1} - \boldsymbol{\theta}^*(\hat{\boldsymbol{\omega}}_t)\|^2$$
$$\leq (1 - \frac{1}{2\kappa})\mathbb{E}\|\boldsymbol{\theta}_t - \boldsymbol{\theta}^*(\hat{\boldsymbol{\omega}}_t)\|^2 + \frac{\eta_{\boldsymbol{\theta}}^2 \sigma^2}{M} \quad (28)$$

Then we return to the inequality (21), we can get:

$$\mathbb{E}\|\boldsymbol{\theta}^*(\hat{\boldsymbol{\omega}}_{t+1}) - \boldsymbol{\theta}_{t+1}\|^2$$
$$\leq (4\kappa - 1)\mathbb{E}\|\boldsymbol{\theta}^*(\hat{\boldsymbol{\omega}}_{t+1}) - \boldsymbol{\theta}^*(\hat{\boldsymbol{\omega}}_t)\|^2$$
$$+ (1 + \frac{1}{4\kappa - 2})(1 - \frac{1}{2\kappa})\mathbb{E}\|\boldsymbol{\theta}_t - \boldsymbol{\theta}^*(\hat{\boldsymbol{\omega}}_t)\|^2$$
$$+ (1 + \frac{1}{4\kappa - 2})\frac{\eta_{\boldsymbol{\theta}}^2 \sigma^2}{M}$$
$$\overset{(i)}{\leq} (1 - \frac{1}{4\kappa})\mathbb{E}\|\boldsymbol{\theta}^*(\hat{\boldsymbol{\omega}}_t) - \boldsymbol{\theta}_t\|^2$$
$$+ (4\kappa - 1)\kappa^2\mathbb{E}\|\hat{\boldsymbol{\omega}}_{t+1} - \hat{\boldsymbol{\omega}}_t\|^2 + \frac{(4\kappa - 1)\eta_{\boldsymbol{\theta}}^2 \sigma^2}{(4\kappa - 2)M}$$
$$\overset{(ii)}{\leq} (1 - \frac{1}{4\kappa})\mathbb{E}\|\boldsymbol{\theta}^*(\boldsymbol{\omega}_t) - \boldsymbol{\theta}_t\|^2$$
$$+ 2(4\kappa - 1)\kappa^2 \eta_{\boldsymbol{\omega}}^2((4\beta^2 l^2 + 2\beta l + 2)\mathbb{E}\|\nabla_{\boldsymbol{\omega}} f(\hat{\boldsymbol{\omega}}_t, \boldsymbol{\theta}_t)\|^2$$
$$+ (5\beta^2 l^2 + 2)\frac{\sigma^2}{M}) + \frac{(4\kappa - 1)\eta_{\boldsymbol{\theta}}^2 \sigma^2}{(4\kappa - 2)M}$$
$$+ 2(4\kappa - 1)\kappa^2 d\Delta^2$$
$$\overset{(iii)}{\leq} [1 - \frac{1}{4\kappa} + 4(4\kappa - 1)\kappa^2 l^2 \eta_{\boldsymbol{\omega}}^2(4\beta^2 l^2 + 2\beta l + 2)]$$
$$\cdot \mathbb{E}\|\boldsymbol{\theta}^*(\hat{\boldsymbol{\omega}}_t) - \boldsymbol{\theta}_t\|^2$$
$$+ 4(4\kappa - 1)\kappa^2 \eta_{\boldsymbol{\omega}}^2(4\beta^2 l^2 + 2\beta l + 2)\mathbb{E}\|\nabla\Phi(\hat{\boldsymbol{\omega}}_t)\|^2$$
$$+ ((4\kappa - 1)\kappa^2 \eta_{\boldsymbol{\omega}}^2(5\beta^2 l^2 + 2) + \frac{4\kappa - 1}{4\kappa - 2}\eta_{\boldsymbol{\theta}}^2)\frac{\sigma^2}{M}$$
$$+ 2(4\kappa - 1)\kappa^2 d\Delta^2$$
(29)

where inequality $(i)$ is due to the $\kappa$-Lipschitz of $\boldsymbol{\theta}^*(\cdot)$ according to the Lemma 4.3 in (Lin et al., 2020); inequality $(ii)$ follows from the decomposition that $\hat{\boldsymbol{\omega}}_{t+1} - \hat{\boldsymbol{\omega}}_t = \hat{\boldsymbol{\omega}}_{t+1} - \boldsymbol{\omega}_{t+1} + \boldsymbol{\omega}_{t+1} - \hat{\boldsymbol{\omega}}_t$ and the update rule of parameter $\boldsymbol{\omega}$, together with Lemma 4.1,A.2; finally inequality $(iii)$ is decomposed by $\|\nabla_{\boldsymbol{\omega}} f(\hat{\boldsymbol{\omega}}_t, \boldsymbol{\theta}_t)\|^2 \leq 2\|\nabla_{\boldsymbol{\omega}} f(\hat{\boldsymbol{\omega}}_t, \boldsymbol{\theta}_t) - \nabla\Phi(\hat{\boldsymbol{\omega}}_t)\|^2 + 2\|\nabla\Phi(\hat{\boldsymbol{\omega}}_t)\|^2 \leq 2l^2\|\boldsymbol{\theta}^*(\hat{\boldsymbol{\omega}}_t) - \boldsymbol{\theta}_t\|^2 + 2\|\nabla\Phi(\hat{\boldsymbol{\omega}}_t)\|^2$ (the second inequality here uses the

property that $\nabla\Phi(\cdot) = \nabla_{\boldsymbol{\omega}} f(\cdot, \boldsymbol{\theta}^*(\cdot))$ which we refer to Lemma 4.3 in (Lin et al., 2020)).

We then simplify the recurrence relation of $\|\boldsymbol{\theta}^*(\hat{\boldsymbol{\omega}}_{t+1}) - \boldsymbol{\theta}_{t+1}\|^2$. We denote the coefficient of $\mathbb{E}\|\boldsymbol{\theta}^*(\hat{\boldsymbol{\omega}}_t) - \boldsymbol{\theta}_t\|^2$ as $\lambda$, i.e. $\lambda = 1 - \frac{1}{4\kappa} + 4(4\kappa - 1)\kappa^2 l^2 \eta_{\boldsymbol{\omega}}^2(4\beta^2 l^2 + 2\beta l + 2)$

Then we repeat the recurrence relation and we can get:

$$\mathbb{E}\|\boldsymbol{\theta}^*(\hat{\boldsymbol{\omega}}_{t+1}) - \boldsymbol{\theta}_{t+1}\|^2$$
$$\leq \lambda\mathbb{E}\|\boldsymbol{\theta}^*(\hat{\boldsymbol{\omega}}_t) - \boldsymbol{\theta}_t\|^2$$
$$+ 4(4\kappa - 1)\kappa^2 \eta_{\boldsymbol{\omega}}^2(4\beta^2 l^2 + 2\beta l + 2)\mathbb{E}\|\nabla\Phi(\hat{\boldsymbol{\omega}}_t)\|^2$$
$$+ ((4\kappa - 1)\kappa^2 \eta_{\boldsymbol{\omega}}^2(5\beta^2 l^2 + 2) + \frac{4\kappa - 1}{4\kappa - 2}\eta_{\boldsymbol{\theta}}^2)\frac{\sigma^2}{M}$$
$$+ 2(4\kappa - 1)\kappa^2 d\Delta^2$$
$$\leq \lambda^2\mathbb{E}\|\boldsymbol{\theta}^*(\hat{\boldsymbol{\omega}}_{t-1}) - \boldsymbol{\theta}_{t-1}\|^2$$
$$+ 4(4\kappa - 1)\kappa^2 \eta_{\boldsymbol{\omega}}^2(4\beta^2 l^2 + 2\beta l + 2)$$
$$\cdot [\lambda\mathbb{E}\|\nabla\Phi(\hat{\boldsymbol{\omega}}_{t-1})\|^2 + \mathbb{E}\|\nabla\Phi(\hat{\boldsymbol{\omega}}_t)\|^2]$$
$$+ ((4\kappa - 1)\kappa^2 \eta_{\boldsymbol{\omega}}^2(5\beta^2 l^2 + 2) + \frac{4\kappa - 1}{4\kappa - 2}\eta_{\boldsymbol{\theta}}^2)\frac{\sigma^2}{M}(\lambda + 1)$$
$$+ 2(4\kappa - 1)\kappa^2 d\Delta^2(\lambda + 1)$$
$$\leq ...$$
$$\leq \lambda^{t+1}\mathbb{E}\|\boldsymbol{\theta}^*(\hat{\boldsymbol{\omega}}_0) - \boldsymbol{\theta}_0\|^2$$
$$+ 4(4\kappa - 1)\kappa^2 \eta_{\boldsymbol{\omega}}^2(4\beta^2 l^2 + 2\beta l + 2)\sum_{j=0}^{t}\lambda^{t-j}\mathbb{E}\|\nabla\Phi(\hat{\boldsymbol{\omega}}_j)\|^2$$
$$+ ((4\kappa - 1)\kappa^2 \eta_{\boldsymbol{\omega}}^2(5\beta^2 l^2 + 2) + \frac{4\kappa - 1}{4\kappa - 2}\eta_{\boldsymbol{\theta}}^2)\frac{\sigma^2}{M}\sum_{j=0}^{t}\lambda^j$$
$$+ 2(4\kappa - 1)\kappa^2 d\Delta^2\sum_{j=0}^{t}\lambda^j$$
$$\leq \lambda^{t+1}D^2$$
$$+ 4(4\kappa - 1)\kappa^2 \eta_{\boldsymbol{\omega}}^2(4\beta^2 l^2 + 2\beta l + 2)\sum_{j=0}^{t}\lambda^{t-j}\mathbb{E}\|\nabla\Phi(\hat{\boldsymbol{\omega}}_j)\|^2$$
$$+ ((4\kappa - 1)\kappa^2 \eta_{\boldsymbol{\omega}}^2(5\beta^2 l^2 + 2) + \frac{4\kappa - 1}{4\kappa - 2}\eta_{\boldsymbol{\theta}}^2)\frac{\sigma^2}{M}\sum_{j=0}^{t}\lambda^j$$
$$+ 2(4\kappa - 1)\kappa^2 d\Delta^2\sum_{j=0}^{t}\lambda^j$$
(30)

where the last inequality is due to Assumption 4.3.

$\square$

**Theorem A.1.** *Under Assumption 4.1,4.2,4.3,4.4 and restrictions* $\eta_{\boldsymbol{\omega}}(2\beta l + 1) \leq \frac{1}{16\kappa^2 l}$ *and* $\eta_{\boldsymbol{\theta}} =$

$\frac{1}{2l}, \kappa \geq 2, \beta l \leq \frac{1}{140}$, *when $f$ is $\mu$-strongly-concave the parameter $\boldsymbol{\theta}$, we have:*

$$\frac{1}{T+1}\sum_{t=0}^{T}\mathbb{E}\|\nabla\Phi(\hat{\boldsymbol{\omega}}_t)\|^2$$

$$\leq \frac{\mathbb{E}[\Phi(\hat{\boldsymbol{\omega}}_0)] - \mathbb{E}[\Phi(\hat{\boldsymbol{\omega}}_{T+1})]}{(T+1)\frac{\eta_{\boldsymbol{\omega}}}{96}} + \frac{576\kappa l^2 D^2}{7(T+1)}$$

$$+ (720\kappa^4\eta_{\boldsymbol{\omega}}^2 l^2 + 210\kappa l\eta_{\boldsymbol{\omega}})\frac{\sigma^2}{M} + (96\kappa + \frac{1}{2})\frac{\sigma^2}{M}$$

$$+ (480\kappa^2 l^2 + \frac{192\kappa l}{\eta_{\boldsymbol{\omega}}} + \frac{768}{\eta_{\boldsymbol{\omega}}^2})d\Delta^2$$

$$(31)$$

*Proof.* We then back to Lemma A.3 and try to make use of Lemma A.4:

$$\mathbb{E}[\Phi(\hat{\boldsymbol{\omega}}_{t+1})]$$

$$\leq \mathbb{E}[\Phi(\hat{\boldsymbol{\omega}}_t)]$$

$$-\frac{\eta_{\boldsymbol{\omega}}}{2}(\frac{15}{16}-5\beta l-4\kappa l\eta_{\boldsymbol{\omega}}(4\beta^2 l^2+2\beta l+2))\mathbb{E}\|\nabla\Phi(\hat{\boldsymbol{\omega}}_t)\|^2$$

$$+l^2[\frac{\eta_{\boldsymbol{\omega}}}{2}(1+\frac{1}{2}\beta l)+2\kappa l\eta_{\boldsymbol{\omega}}^2(4\beta^2 l^2+2\beta l+2)]\mathbb{E}[\|\boldsymbol{\theta}^*(\hat{\boldsymbol{\omega}}_t)-$$

$$\boldsymbol{\theta}_t\|^2] + (5\beta^2 l^2 + 2)\frac{\kappa l\eta_{\boldsymbol{\omega}}^2\sigma^2}{M}$$

$$+\frac{\beta l\sigma^2\eta_{\boldsymbol{\omega}}}{M} + (2\kappa l + \frac{8}{\eta_{\boldsymbol{\omega}}})d\Delta^2$$

where we use the property that $\nabla\Phi(\cdot) = \nabla_{\boldsymbol{\omega}}f(\cdot, \boldsymbol{\theta}^*(\cdot))$.

Then we use Lemma A.4, we can get:

$$\mathbb{E}[\Phi(\hat{\boldsymbol{\omega}}_{t+1})]$$

$$\leq \mathbb{E}[\Phi(\hat{\boldsymbol{\omega}}_t)]$$

$$-\frac{\eta_{\boldsymbol{\omega}}}{2}(\frac{15}{16}-5\beta l-4\kappa l\eta_{\boldsymbol{\omega}}(4\beta^2 l^2+2\beta l+2))\mathbb{E}\|\nabla\Phi(\hat{\boldsymbol{\omega}}_t)\|^2$$

$$+\frac{\eta_{\boldsymbol{\omega}}}{2}l^2[1+\frac{1}{2}\beta l+4\kappa l\eta_{\boldsymbol{\omega}}(4\beta^2 l^2+2\beta l+2)]\lambda^t D^2$$

$$+\frac{\eta_{\boldsymbol{\omega}}}{2}l^2[1+\frac{1}{2}\beta l+4\kappa l\eta_{\boldsymbol{\omega}}(4\beta^2 l^2+2\beta l+2)]$$

$$\cdot 4(4\kappa-1)\kappa^2\eta_{\boldsymbol{\omega}}^2(4\beta^2 l^2+2\beta l+2)\sum_{j=0}^{t-1}\lambda^{t-1-j}\mathbb{E}\|\nabla\Phi(\hat{\boldsymbol{\omega}}_j)\|^2$$

$$+\frac{\eta_{\boldsymbol{\omega}}}{2}l^2[1+\frac{1}{2}\beta l+4\kappa l\eta_{\boldsymbol{\omega}}(4\beta^2 l^2+2\beta l+2)]$$

$$\cdot((4\kappa-1)\kappa^2\eta_{\boldsymbol{\omega}}^2(5\beta^2 l^2+2)+\frac{4\kappa-1}{4\kappa-2}\eta_{\boldsymbol{\theta}}^2)\frac{\sigma^2}{M}\sum_{j=0}^{t-1}\lambda^j$$

$$+\frac{\eta_{\boldsymbol{\omega}}}{2}l^2[1+\frac{1}{2}\beta l+4\kappa l\eta_{\boldsymbol{\omega}}(4\beta^2 l^2+2\beta l+2)]$$

$$+2(4\kappa-1)\kappa^2 d\Delta^2\sum_{j=0}^{t}\lambda^j$$

$$+(5\beta^2 l^2+2)\frac{\kappa l\eta_{\boldsymbol{\omega}}^2\sigma^2}{M}+\frac{\beta l\sigma^2\eta_{\boldsymbol{\omega}}}{M}+(2\kappa l+\frac{8}{\eta_{\boldsymbol{\omega}}})d\Delta^2$$

$$(32)$$

We then repeat the inequality (32) above from

$t = T$ to $t = 0$, we have:

$$\mathbb{E}[\Phi(\hat{\boldsymbol{\omega}}_{T+1})]$$
$$\leq \mathbb{E}[\Phi(\hat{\boldsymbol{\omega}}_0)]$$
$$-\frac{\eta_{\boldsymbol{\omega}}}{2}(\frac{15}{16}-5\beta l-4\kappa l\eta_{\boldsymbol{\omega}}(4\beta^2l^2+2\beta l+2))\sum_{t=0}^{T}\mathbb{E}\|\nabla\Phi(\hat{\boldsymbol{\omega}}_t)\|^2$$
$$+\frac{\eta_{\boldsymbol{\omega}}}{2}l^2[1+\frac{1}{2}\beta l+4\kappa l\eta_{\boldsymbol{\omega}}(4\beta^2l^2+2\beta l+2)]D^2\sum_{t=0}^{T}\lambda^t$$
$$+\frac{\eta_{\boldsymbol{\omega}}}{2}l^2[1+\frac{1}{2}\beta l+4\kappa l\eta_{\boldsymbol{\omega}}(4\beta^2l^2+2\beta l+2)]$$
$$\cdot 4(4\kappa-1)\kappa^2\eta_{\boldsymbol{\omega}}^2(4\beta^2l^2+2\beta l+2)$$
$$\cdot\sum_{t=0}^{T-1}\sum_{j=0}^{t-1}\lambda^{t-1-j}\mathbb{E}\|\nabla\Phi(\hat{\boldsymbol{\omega}}_j)\|^2$$
$$+\frac{\eta_{\boldsymbol{\omega}}}{2}l^2[1+\frac{1}{2}\beta l+4\kappa l\eta_{\boldsymbol{\omega}}(4\beta^2l^2+2\beta l+2)]$$
$$\cdot((4\kappa-1)\kappa^2\eta_{\boldsymbol{\omega}}^2(5\beta^2l^2+2)+\frac{4\kappa-1}{4\kappa-2}\eta_{\boldsymbol{\theta}}^2)$$
$$\cdot\frac{\sigma^2}{M}\sum_{t=0}^{T-1}\sum_{j=0}^{t-1}\lambda^j$$
$$+\frac{\eta_{\boldsymbol{\omega}}}{2}l^2[1+\frac{1}{2}\beta l+4\kappa l\eta_{\boldsymbol{\omega}}(4\beta^2l^2+2\beta l+2)]$$
$$\cdot 2(4\kappa-1)\kappa^2d\Delta^2\sum_{t=0}^{T-1}\sum_{j=0}^{t-1}\lambda^j$$
$$+((5\beta^2l^2+2)\frac{\kappa l\eta_{\boldsymbol{\omega}}^2\sigma^2}{M}+\frac{\beta l\sigma^2\eta_{\boldsymbol{\omega}}}{M}+(2\kappa l+\frac{8}{\eta_{\boldsymbol{\omega}}})d\Delta^2)(T+1)$$
$$\tag{33}$$

Since the parameters satisfy: $\eta_{\boldsymbol{\omega}}(2\beta l+1)\leq\frac{1}{16\kappa^2l}$, we can evaluate that $\lambda = 1-\frac{1}{4\kappa}+2(4\kappa-1)\kappa^2l^2\eta_{\boldsymbol{\omega}}^2(4\beta^2l^2+2\beta l+2)\leq 1-\frac{1}{4\kappa}+\frac{1}{16\kappa}=1-\frac{3}{16\kappa}$. Therefore the summation can be bounded:$\sum_{t=0}^{T}\lambda^t\leq\sum_{t=0}^{T}(1-\frac{3}{16\kappa})^t\leq\frac{16\kappa}{3}$.

So the above inequality can be turned to:

$$\mathbb{E}[\Phi(\hat{\boldsymbol{\omega}}_{T+1})]$$
$$\leq\mathbb{E}[\Phi(\hat{\boldsymbol{\omega}}_0)]$$
$$-\frac{\eta_{\boldsymbol{\omega}}}{2}(\frac{15}{16}-5\beta l-4\kappa l\eta_{\boldsymbol{\omega}}(4\beta^2l^2+2\beta l+2))\sum_{t=0}^{T}\mathbb{E}\|\nabla\Phi(\hat{\boldsymbol{\omega}}_t)\|^2$$
$$+\frac{\eta_{\boldsymbol{\omega}}}{2}l^2[1+\frac{1}{2}\beta l+4\kappa l\eta_{\boldsymbol{\omega}}(4\beta^2l^2+2\beta l+2)]D^2\frac{16\kappa}{3}$$
$$+\frac{\eta_{\boldsymbol{\omega}}}{2}l^2[1+\frac{1}{2}\beta l+4\kappa l\eta_{\boldsymbol{\omega}}(4\beta^2l^2+2\beta l+2)]$$
$$\cdot 4(4\kappa-1)\kappa^2\eta_{\boldsymbol{\omega}}^2(4\beta^2l^2+2\beta l+2)\frac{16\kappa}{3}\sum_{t=0}^{T}\mathbb{E}\|\nabla\Phi(\hat{\boldsymbol{\omega}}_t)\|^2$$
$$+\frac{\eta_{\boldsymbol{\omega}}}{2}l^2[1+\frac{1}{2}\beta l+4\kappa l\eta_{\boldsymbol{\omega}}(4\beta^2l^2+2\beta l+2)]$$

$$\cdot((4\kappa-1)\kappa^2\eta_{\boldsymbol{\omega}}^2(5\beta^2l^2+2)+\frac{4\kappa-1}{4\kappa-2}\eta_{\boldsymbol{\theta}}^2)\frac{\sigma^2}{M}\frac{16\kappa}{3}(T+1)$$
$$+\frac{\eta_{\boldsymbol{\omega}}}{2}l^2[1+\frac{1}{2}\beta l+4\kappa l\eta_{\boldsymbol{\omega}}(4\beta^2l^2+2\beta l+2)]$$
$$\cdot 2(4\kappa-1)\kappa^2d\Delta^2\frac{16\kappa}{3}(T+1)$$
$$+((5\beta^2l^2+2)\frac{\kappa l\eta_{\boldsymbol{\omega}}^2\sigma^2}{M}+\frac{\beta l\sigma^2\eta_{\boldsymbol{\omega}}}{M}+(2\kappa l+\frac{8}{\eta_{\boldsymbol{\omega}}})d\Delta^2)(T+1)$$
$$\leq\mathbb{E}[\Phi(\hat{\boldsymbol{\omega}}_0)]$$
$$-\frac{\eta_{\boldsymbol{\omega}}}{2}(\frac{15}{16}-5\beta l-4\kappa l\eta_{\boldsymbol{\omega}}(4\beta^2l^2+2\beta l+2))\sum_{t=0}^{T}\mathbb{E}\|\nabla\Phi(\hat{\boldsymbol{\omega}}_t)\|^2$$
$$+\frac{\eta_{\boldsymbol{\omega}}}{2}l^2[1+\frac{1}{2}\beta l+4\kappa l\eta_{\boldsymbol{\omega}}(4\beta^2l^2+2\beta l+2)]D^2\frac{16\kappa}{3}$$
$$+\frac{\eta_{\boldsymbol{\omega}}}{6}[1+\frac{1}{2}\beta l+4\kappa l\eta_{\boldsymbol{\omega}}(4\beta^2l^2+2\beta l+2)]$$
$$\cdot\sum_{t=0}^{T}\mathbb{E}\|\nabla\Phi(\hat{\boldsymbol{\omega}}_t)\|^2$$
$$+\frac{\eta_{\boldsymbol{\omega}}}{2}l^2[1+\frac{1}{2}\beta l+4\kappa l\eta_{\boldsymbol{\omega}}(4\beta^2l^2+2\beta l+2)]$$
$$\cdot 2(4\kappa-1)\kappa^2d\Delta^2\frac{16\kappa}{3}(T+1)$$
$$+((5\beta^2l^2+2)\frac{\kappa l\eta_{\boldsymbol{\omega}}^2\sigma^2}{M}+\frac{\beta l\sigma^2\eta_{\boldsymbol{\omega}}}{M}+(2\kappa l+\frac{8}{\eta_{\boldsymbol{\omega}}})d\Delta^2)(T+1)$$
$$\leq\mathbb{E}[\Phi(\hat{\boldsymbol{\omega}}_0)]$$
$$-\frac{\eta_{\boldsymbol{\omega}}}{6}(\frac{13}{16}-16\beta l-20\kappa l\eta_{\boldsymbol{\omega}}(4\beta^2l^2+2\beta l+2))$$
$$\cdot\sum_{t=0}^{T}\mathbb{E}\|\nabla\Phi(\hat{\boldsymbol{\omega}}_t)\|^2$$
$$+\frac{8\kappa\eta_{\boldsymbol{\omega}}}{3}l^2D^2[1+\frac{1}{2}\beta l+4\kappa l\eta_{\boldsymbol{\omega}}(4\beta^2l^2+2\beta l+2)]$$
$$+\frac{8\kappa\eta_{\boldsymbol{\omega}}}{3}l^2[1+\frac{1}{2}\beta l+4\kappa l\eta_{\boldsymbol{\omega}}(4\beta^2l^2+2\beta l+2)]$$
$$\cdot((4\kappa-1)\kappa^2\eta_{\boldsymbol{\omega}}^2(5\beta^2l^2+2)+\frac{4\kappa-1}{4\kappa-2}\eta_{\boldsymbol{\theta}}^2)\frac{\sigma^2}{M}(T+1)$$
$$+\frac{16\kappa\eta_{\boldsymbol{\omega}}}{3}l^2[1+\frac{1}{2}\beta l+4\kappa l\eta_{\boldsymbol{\omega}}(4\beta^2l^2+2\beta l+2)]$$
$$\cdot(4\kappa-1)\kappa^2d\Delta^2(T+1)$$
$$+((5\beta^2l^2+2)\frac{\kappa l\eta_{\boldsymbol{\omega}}^2\sigma^2}{M}+\frac{\beta l\sigma^2\eta_{\boldsymbol{\omega}}}{M}+(2\kappa l+\frac{8}{\eta_{\boldsymbol{\omega}}})d\Delta^2)(T+1)$$
$$\leq\mathbb{E}[\Phi(\hat{\boldsymbol{\omega}}_0)]-\frac{\eta_{\boldsymbol{\omega}}}{96}\sum_{t=0}^{T}\mathbb{E}\|\nabla\Phi(\hat{\boldsymbol{\omega}}_t)\|^2+\frac{6}{7}\kappa\eta_{\boldsymbol{\omega}}l^2D^2$$
$$+(\frac{360}{49}\kappa^4\eta_{\boldsymbol{\omega}}^3l^2+\kappa\eta_{\boldsymbol{\omega}})\frac{\sigma^2}{M}(T+1)+\frac{240}{49}\kappa^2\eta_{\boldsymbol{\omega}}l^2d\Delta^2(T+1)$$
$$+(\frac{15}{7}\frac{\kappa l\eta_{\boldsymbol{\omega}}^2\sigma^2}{M}+\frac{\eta_{\boldsymbol{\omega}}}{140}\frac{\sigma^2}{M}+(2\kappa l+\frac{8}{\eta_{\boldsymbol{\omega}}})d\Delta^2)(T+1)$$

on the condition that: $\kappa\geq 2, \beta l\leq\frac{1}{140}$ So we can get the average sum of $\mathbb{E}\|\nabla\Phi(\hat{\boldsymbol{\omega}}_t)\|^2$ bounded

by:

$$\frac{1}{T+1}\sum_{t=0}^{T}\mathbb{E}\|\nabla\Phi(\hat{\boldsymbol{\omega}}_t)\|^2$$

$$\leq \frac{\mathbb{E}[\Phi(\hat{\boldsymbol{\omega}}_0)]-\mathbb{E}[\Phi(\hat{\boldsymbol{\omega}}_{T+1})]}{(T+1)\frac{\eta_{\boldsymbol{\omega}}}{96}}+\frac{576\kappa l^2 D^2}{7(T+1)}$$

$$+(720\kappa^4\eta_{\boldsymbol{\omega}}^2 l^2+210\kappa l\eta_{\boldsymbol{\omega}})\frac{\sigma^2}{M}+(96\kappa+\frac{1}{2})\frac{\sigma^2}{M}$$

$$+(480\kappa^2 l^2+\frac{192\kappa l}{\eta_{\boldsymbol{\omega}}}+\frac{768}{\eta_{\boldsymbol{\omega}}^2})d\Delta^2 \tag{34}$$

So the bound for the algorithm to get an $\epsilon$-stationary point is

$$\mathcal{O}(\frac{\kappa^2\Delta_\Phi+\kappa D^2}{\epsilon^2}\max\{1,\frac{\kappa\sigma^2}{\epsilon^2}\})$$

$\square$

### A.6 PL Condition

First we construct a potential function in the same way as (Yang et al., 2022):

$$V_t = V(\hat{\boldsymbol{\omega}}_t,\boldsymbol{\theta}_t)=\Phi(\hat{\boldsymbol{\omega}}_t)+\alpha[\Phi(\hat{\boldsymbol{\omega}}_t)-f(\hat{\boldsymbol{\omega}}_t,\boldsymbol{\theta}_t)]$$

where $\alpha>0$ is a preset parameter. Then we come to evaluate the descending relationship of the potential function $V_t$.

**Theorem A.2.** *Under Assumption 4.1,4.2,4.3,4.4 and restrictions* $\alpha=\frac{1}{16}$, $\beta l\leq\frac{1}{16}$, $\eta_{\boldsymbol{\omega}}(2\beta l+1)^2\kappa l\leq\frac{1}{64}$, $\kappa^2\eta_{\boldsymbol{\omega}}l\leq\frac{1}{128}$ *and* $\eta_{\boldsymbol{\theta}}=64\kappa^2\eta_{\boldsymbol{\omega}}$, *when $f$ satisfies $\mu$-PL condition on parameter $\boldsymbol{\theta}$, we have:*

$$\frac{1}{T}\sum_{t=0}^{T-1}\mathbb{E}\|\nabla\Phi(\hat{\boldsymbol{\omega}}_t)\|^2$$

$$\leq\mathcal{O}(\frac{\Phi(\hat{\boldsymbol{\omega}}_0)-\Phi^*}{\eta_{\boldsymbol{\omega}}T})+\mathcal{O}(\frac{\eta_{\boldsymbol{\omega}}\kappa^4\sigma^2}{M})+\mathcal{O}((\eta_{\boldsymbol{\theta}}+\frac{1}{\eta_{\boldsymbol{\omega}}})d\Delta^2) \tag{35}$$

*Proof.* We get the conclusion that $\Phi(\boldsymbol{\omega})$ is $L$-smooth according to Lemma A.5 in (Nouiehed et al., 2019), where $L=l+\frac{l\kappa}{2}$. And we can get the descending relationship of $\mathbb{E}[\Phi(\hat{\boldsymbol{\omega}}_t)]$ in the

same way as Lemma A.3:

$$\mathbb{E}[\Phi(\hat{\boldsymbol{\omega}}_{t+1})]$$

$$\leq\mathbb{E}[\Phi(\hat{\boldsymbol{\omega}}_t)]$$

$$-\frac{\eta_{\boldsymbol{\omega}}}{2}(\frac{15}{16}-5\beta l-2L\eta_{\boldsymbol{\omega}}(4\beta^2 l^2+2\beta l+2))\mathbb{E}\|\nabla\Phi(\hat{\boldsymbol{\omega}}_t)\|^2$$

$$+[\frac{\eta_{\boldsymbol{\omega}}}{2}(1+\frac{1}{2}\beta l)+L\eta_{\boldsymbol{\omega}}^2(4\beta^2 l^2+2\beta l+2)]\mathbb{E}[\|\nabla\Phi(\hat{\boldsymbol{\omega}}_t)-$$

$$\nabla_{\boldsymbol{\omega}}f(\hat{\boldsymbol{\omega}}_t,\boldsymbol{\theta}_t)\|^2]+(5\beta^2 l^2+2)\frac{L\eta_{\boldsymbol{\omega}}^2\sigma^2}{2M}$$

$$+\frac{\beta l\sigma^2\eta_{\boldsymbol{\omega}}}{M}+(L+\frac{8}{\eta_{\boldsymbol{\omega}}})d\Delta^2 \tag{36}$$

And then using the smoothness of the variables $\boldsymbol{\omega}$ and $\boldsymbol{\theta}$ respectively, we can get:

$$f(\hat{\boldsymbol{\omega}}_{t+1},\boldsymbol{\theta}_t)-f(\hat{\boldsymbol{\omega}}_t,\boldsymbol{\theta}_t)$$

$$\geq\langle\nabla_{\boldsymbol{\omega}}f(\hat{\boldsymbol{\omega}}_t,\boldsymbol{\theta}_t),\hat{\boldsymbol{\omega}}_{t+1}-\hat{\boldsymbol{\omega}}_t\rangle-\frac{l}{2}\|\hat{\boldsymbol{\omega}}_{t+1}-\hat{\boldsymbol{\omega}}_t\|^2$$

$$f(\hat{\boldsymbol{\omega}}_{t+1},\boldsymbol{\theta}_{t+1})-f(\hat{\boldsymbol{\omega}}_{t+1},\boldsymbol{\theta}_t)$$

$$\geq\langle\nabla_{\boldsymbol{\theta}}f(\hat{\boldsymbol{\omega}}_{t+1},\boldsymbol{\theta}_t),\boldsymbol{\theta}_{t+1}-\boldsymbol{\theta}_t\rangle-\frac{l}{2}\|\boldsymbol{\theta}_{t+1}-\boldsymbol{\theta}_t\|^2$$

Taking expectations respectively we can get:

$$\mathbb{E}[f(\hat{\boldsymbol{\omega}}_{t+1},\boldsymbol{\theta}_t)]$$

$$\geq\mathbb{E}[f(\hat{\boldsymbol{\omega}}_t,\boldsymbol{\theta}_t)]-\eta_{\boldsymbol{\omega}}\mathbb{E}\langle\nabla_{\boldsymbol{\omega}}f(\hat{\boldsymbol{\omega}}_t,\boldsymbol{\theta}_t),\nabla_{\boldsymbol{\omega}}f(\widetilde{\boldsymbol{\omega}}_{t+1/2},\boldsymbol{\theta}_t)\rangle$$

$$-l\eta_{\boldsymbol{\omega}}^2\mathbb{E}\|g_{\boldsymbol{\omega}}(\widetilde{\boldsymbol{\omega}}_{t+1/2},\boldsymbol{\theta}_t)\|^2-2ld\Delta^2$$

$$-\frac{\eta_{\boldsymbol{\omega}}}{32}\mathbb{E}\|\nabla_{\boldsymbol{\omega}}f(\hat{\boldsymbol{\omega}}_t,\boldsymbol{\theta}_t)\|^2-\frac{8}{\eta_{\boldsymbol{\omega}}}d\Delta^2$$

$$\geq\mathbb{E}[f(\hat{\boldsymbol{\omega}}_t,\boldsymbol{\theta}_t)]-\eta_{\boldsymbol{\omega}}\mathbb{E}\|\nabla_{\boldsymbol{\omega}}f(\hat{\boldsymbol{\omega}}_t,\boldsymbol{\theta}_t)\|^2-\frac{\eta_{\boldsymbol{\omega}}}{2}\mathbb{E}\|\nabla_{\boldsymbol{\omega}}f(\hat{\boldsymbol{\omega}}_t,\boldsymbol{\theta}_t)\|^2$$

$$-\frac{\eta_{\boldsymbol{\omega}}}{2}\mathbb{E}\|\nabla_{\boldsymbol{\omega}}f(\widetilde{\boldsymbol{\omega}}_{t+1/2},\boldsymbol{\theta}_t)-\nabla_{\boldsymbol{\omega}}f(\hat{\boldsymbol{\omega}}_t,\boldsymbol{\theta}_t)\|^2$$

$$-l\eta_{\boldsymbol{\omega}}^2\mathbb{E}\|g_{\boldsymbol{\omega}}(\widetilde{\boldsymbol{\omega}}_{t+1/2},\boldsymbol{\theta}_t)\|^2-2ld\Delta^2$$

$$-\frac{\eta_{\boldsymbol{\omega}}}{32}\mathbb{E}\|\nabla_{\boldsymbol{\omega}}f(\hat{\boldsymbol{\omega}}_t,\boldsymbol{\theta}_t)\|^2-\frac{8}{\eta_{\boldsymbol{\omega}}}d\Delta^2$$

$$\geq\mathbb{E}[f(\hat{\boldsymbol{\omega}}_t,\boldsymbol{\theta}_t)]-\frac{49\eta_{\boldsymbol{\omega}}}{32}\mathbb{E}\|\nabla_{\boldsymbol{\omega}}f(\hat{\boldsymbol{\omega}}_t,\boldsymbol{\theta}_t)\|^2$$

$$-\frac{l^2\beta^2\eta_{\boldsymbol{\omega}}}{2}\mathbb{E}\|g_{\boldsymbol{\omega}}(\hat{\boldsymbol{\omega}}_t,\boldsymbol{\theta}_t)\|^2-l\eta_{\boldsymbol{\omega}}^2\mathbb{E}\|g_{\boldsymbol{\omega}}(\widetilde{\boldsymbol{\omega}}_{t+1/2},\boldsymbol{\theta}_t)\|^2$$

$$-(2l+\frac{8}{\eta_{\boldsymbol{\omega}}})d\Delta^2$$

$$\geq\mathbb{E}[f(\hat{\boldsymbol{\omega}}_t,\boldsymbol{\theta}_t)]-(\frac{49\eta_{\boldsymbol{\omega}}}{32}+\frac{l^2\beta^2\eta_{\boldsymbol{\omega}}}{2}+l\eta_{\boldsymbol{\omega}}^2(4\beta^2 l^2+2\beta l+2))$$

$$\cdot\mathbb{E}\|\nabla_{\boldsymbol{\omega}}f(\hat{\boldsymbol{\omega}}_t,\boldsymbol{\theta}_t)\|^2-(\frac{l^2\beta^2\eta_{\boldsymbol{\omega}}}{2}+l\eta_{\boldsymbol{\omega}}^2(5\beta^2 l^2+2))\frac{\sigma^2}{M}$$

$$-(2l+\frac{8}{\eta_{\boldsymbol{\omega}}})d\Delta^2 \tag{37}$$

$$\mathbb{E}[f(\hat{\boldsymbol{\omega}}_{t+1}, \boldsymbol{\theta}_{t+1})]$$
$$\geq \mathbb{E}[f(\hat{\boldsymbol{\omega}}_{t+1}, \boldsymbol{\theta}_t)] + \eta_{\boldsymbol{\theta}}\mathbb{E}\langle \nabla_{\boldsymbol{\theta}} f(\hat{\boldsymbol{\omega}}_{t+1}, \boldsymbol{\theta}_t), \nabla_{\boldsymbol{\theta}} f(\hat{\boldsymbol{\omega}}_t, \boldsymbol{\theta}_t)\rangle$$
$$- \frac{l\eta_{\boldsymbol{\theta}}^2}{2}\mathbb{E}\|g_{\boldsymbol{\theta}}(\hat{\boldsymbol{\omega}}_t, \boldsymbol{\theta}_t)\|^2$$
$$\geq \mathbb{E}[f(\hat{\boldsymbol{\omega}}_{t+1}, \boldsymbol{\theta}_t)] + \frac{\eta_{\boldsymbol{\theta}}}{2}\mathbb{E}\|\nabla_{\boldsymbol{\theta}} f(\hat{\boldsymbol{\omega}}_t, \boldsymbol{\theta}_t)\|^2$$
$$- \frac{\eta_{\boldsymbol{\theta}}}{2}\mathbb{E}\|\nabla_{\boldsymbol{\theta}} f(\hat{\boldsymbol{\omega}}_{t+1}, \boldsymbol{\theta}_t) - \nabla_{\boldsymbol{\theta}} f(\hat{\boldsymbol{\omega}}_t, \boldsymbol{\theta}_t)\|^2$$
$$- \frac{l\eta_{\boldsymbol{\theta}}^2}{2}\mathbb{E}\|g_{\boldsymbol{\theta}}(\hat{\boldsymbol{\omega}}_t, \boldsymbol{\theta}_t)\|^2$$
$$\geq \mathbb{E}[f(\hat{\boldsymbol{\omega}}_{t+1}, \boldsymbol{\theta}_t)] + \frac{\eta_{\boldsymbol{\theta}}}{2}\mathbb{E}\|\nabla_{\boldsymbol{\theta}} f(\hat{\boldsymbol{\omega}}_t, \boldsymbol{\theta}_t)\|^2$$
$$- l^2\eta_{\boldsymbol{\omega}}^2\eta_{\boldsymbol{\theta}}\mathbb{E}\|g_{\boldsymbol{\omega}}(\widetilde{\boldsymbol{\omega}}_{t+1/2}, \boldsymbol{\theta}_t)\|^2 - \frac{l\eta_{\boldsymbol{\theta}}^2}{2}\mathbb{E}\|g_{\boldsymbol{\theta}}(\hat{\boldsymbol{\omega}}_t, \boldsymbol{\theta}_t)\|^2$$
$$- l^2\eta_{\boldsymbol{\theta}} d\Delta^2$$
$$\geq \mathbb{E}[f(\hat{\boldsymbol{\omega}}_{t+1}, \boldsymbol{\theta}_t)] + (\frac{\eta_{\boldsymbol{\theta}}}{2} - \frac{l\eta_{\boldsymbol{\theta}}^2}{2})\mathbb{E}\|\nabla_{\boldsymbol{\theta}} f(\hat{\boldsymbol{\omega}}_t, \boldsymbol{\theta}_t)\|^2$$
$$- (l^2\eta_{\boldsymbol{\omega}}^2\eta_{\boldsymbol{\theta}}(4\beta^2l^2 + 2\beta l + 2))\mathbb{E}\|\nabla_{\boldsymbol{\omega}} f(\hat{\boldsymbol{\omega}}_t, \boldsymbol{\theta}_t)\|^2$$
$$- (\frac{l\eta_{\boldsymbol{\theta}}^2}{2} + l^2\eta_{\boldsymbol{\omega}}^2\eta_{\boldsymbol{\theta}}(5\beta^2l^2 + 2))\frac{\sigma^2}{M} - l^2\eta_{\boldsymbol{\theta}} d\Delta^2$$
$$\tag{38}$$

Combining the above inequalities we can get the descending relationship of the potential function:

$$\mathbb{E}[V_{t+1}] - \mathbb{E}[V_t]$$
$$= (1 + \alpha)(\mathbb{E}[\Phi(\hat{\boldsymbol{\omega}}_{t+1})] - \mathbb{E}[\Phi(\hat{\boldsymbol{\omega}}_t)])$$
$$- \alpha(\mathbb{E}[f(\hat{\boldsymbol{\omega}}_{t+1}, \boldsymbol{\theta}_{t+1})] - \mathbb{E}[f(\hat{\boldsymbol{\omega}}_t, \boldsymbol{\theta}_t)])$$
$$\leq (1+\alpha)\{-\frac{\eta_{\boldsymbol{\omega}}}{2}(\frac{15}{16} - 5\beta l - 2L\eta_{\boldsymbol{\omega}}(4\beta^2l^2 + 2\beta l + 2))$$
$$\cdot \mathbb{E}\|\nabla\Phi(\hat{\boldsymbol{\omega}}_t)\|^2 + [\frac{\eta_{\boldsymbol{\omega}}}{2}(1 + \frac{1}{2}\beta l) + L\eta_{\boldsymbol{\omega}}^2(4\beta^2l^2 + 2\beta l + 2)]$$
$$\cdot \mathbb{E}[\|\nabla\Phi(\hat{\boldsymbol{\omega}}_t) - \nabla_{\boldsymbol{\omega}} f(\hat{\boldsymbol{\omega}}_t, \boldsymbol{\theta}_t)\|^2] + (5\beta^2l^2 + 2)\frac{L\eta_{\boldsymbol{\omega}}^2\sigma^2}{2M}$$
$$+ \frac{\beta l\sigma^2\eta_{\boldsymbol{\omega}}}{M} + (L + \frac{8}{\eta_{\boldsymbol{\omega}}})d\Delta^2\}$$
$$-\alpha\{-(\frac{49\eta_{\boldsymbol{\omega}}}{32} + \frac{l^2\beta^2\eta_{\boldsymbol{\omega}}}{2} + l\eta_{\boldsymbol{\omega}}^2(4\beta^2l^2 + 2\beta l + 2))$$
$$\cdot \mathbb{E}\|\nabla_{\boldsymbol{\omega}} f(\hat{\boldsymbol{\omega}}_t, \boldsymbol{\theta}_t)\|^2 - (\frac{l^2\beta^2\eta_{\boldsymbol{\omega}}}{2} + l\eta_{\boldsymbol{\omega}}^2(5\beta^2l^2 + 2))\frac{\sigma^2}{M}$$
$$- (2l + \frac{8}{\eta_{\boldsymbol{\omega}}})d\Delta^2$$
$$+ (\frac{\eta_{\boldsymbol{\theta}}}{2} - \frac{l\eta_{\boldsymbol{\theta}}^2}{2})\mathbb{E}\|\nabla_{\boldsymbol{\theta}} f(\hat{\boldsymbol{\omega}}_t, \boldsymbol{\theta}_t)\|^2$$
$$- (l^2\eta_{\boldsymbol{\omega}}^2\eta_{\boldsymbol{\theta}}(4\beta^2l^2 + 2\beta l + 2))\mathbb{E}\|\nabla_{\boldsymbol{\omega}} f(\hat{\boldsymbol{\omega}}_t, \boldsymbol{\theta}_t)\|^2$$
$$- (\frac{l\eta_{\boldsymbol{\theta}}^2}{2} + l^2\eta_{\boldsymbol{\omega}}^2\eta_{\boldsymbol{\theta}}(5\beta^2l^2 + 2))\frac{\sigma^2}{M} - l^2\eta_{\boldsymbol{\theta}} d\Delta^2\}$$
$$= -\frac{\eta_{\boldsymbol{\omega}}}{2}(1+\alpha)(\frac{15}{16} - 5\beta l - 2L\eta_{\boldsymbol{\omega}}(4\beta^2l^2 + 2\beta l + 2))$$
$$\cdot \mathbb{E}\|\nabla\Phi(\hat{\boldsymbol{\omega}}_t)\|^2 + (1+\alpha)(\frac{\eta_{\boldsymbol{\omega}}}{2}(1+\frac{1}{2}\beta l) + L\eta_{\boldsymbol{\omega}}^2(4\beta^2l^2 + 2\beta l + 2))$$

$$\cdot \mathbb{E}\|\nabla\Phi(\hat{\boldsymbol{\omega}}_t) - \nabla_{\boldsymbol{\omega}} f(\hat{\boldsymbol{\omega}}_t, \boldsymbol{\theta}_t)\|^2$$
$$+ \alpha[(\frac{49\eta_{\boldsymbol{\omega}}}{32} + \frac{l^2\beta^2\eta_{\boldsymbol{\omega}}}{2} + l\eta_{\boldsymbol{\omega}}^2(4\beta^2l^2 + 2\beta l + 2))$$
$$+ l^2\eta_{\boldsymbol{\omega}}^2\eta_{\boldsymbol{\theta}}(4\beta^2l^2 + 2\beta l + 2)]\mathbb{E}\|\nabla_{\boldsymbol{\omega}} f(\hat{\boldsymbol{\omega}}_t, \boldsymbol{\theta}_t)\|^2$$
$$- \alpha(\frac{\eta_{\boldsymbol{\theta}}}{2} - \frac{l\eta_{\boldsymbol{\theta}}^2}{2})\mathbb{E}\|\nabla_{\boldsymbol{\theta}} f(\hat{\boldsymbol{\omega}}_t, \boldsymbol{\theta}_t)\|^2$$
$$+ [(1+\alpha)(5\beta^2l^2 + 2)\frac{L\eta_{\boldsymbol{\omega}}^2}{2} + \alpha(\frac{l^2\beta^2\eta_{\boldsymbol{\omega}}}{2} + l\eta_{\boldsymbol{\omega}}^2(5\beta^2l^2 + 2))$$
$$+ \alpha(\frac{l\eta_{\boldsymbol{\theta}}^2}{2} + \frac{l^2\eta_{\boldsymbol{\omega}}^2\eta_{\boldsymbol{\theta}}}{2}(5\beta^2l^2 + 2))]\frac{\sigma^2}{M}$$
$$+ ((1+\alpha)(L + \frac{8}{\eta_{\boldsymbol{\omega}}}) + \alpha(2l + \frac{8}{\eta_{\boldsymbol{\omega}}}) + \alpha l^2\eta_{\boldsymbol{\theta}})d\Delta^2$$
$$\leq -\{\frac{\eta_{\boldsymbol{\omega}}}{2}(1+\alpha)(\frac{15}{16} - 5\beta l - 2L\eta_{\boldsymbol{\omega}}(4\beta^2l^2 + 2\beta l + 2))$$
$$- 2\alpha[(\frac{49\eta_{\boldsymbol{\omega}}}{32} + \frac{l^2\beta^2\eta_{\boldsymbol{\omega}}}{2} + l\eta_{\boldsymbol{\omega}}^2(4\beta^2l^2 + 2\beta l + 2))$$
$$+ l^2\eta_{\boldsymbol{\omega}}^2\eta_{\boldsymbol{\theta}}(4\beta^2l^2 + 2\beta l + 2)]\}\mathbb{E}\|\nabla\Phi(\hat{\boldsymbol{\omega}}_t)\|^2$$
$$+ \{(1+\alpha)(\frac{\eta_{\boldsymbol{\omega}}}{2}(1 + \frac{1}{2}\beta l) + L\eta_{\boldsymbol{\omega}}^2(4\beta^2l^2 + 2\beta l + 2))$$
$$+ 2\alpha[(\frac{49\eta_{\boldsymbol{\omega}}}{32} + \frac{l^2\beta^2\eta_{\boldsymbol{\omega}}}{2} + l\eta_{\boldsymbol{\omega}}^2(4\beta^2l^2 + 2\beta l + 2))$$
$$+ l^2\eta_{\boldsymbol{\omega}}^2\eta_{\boldsymbol{\theta}}(4\beta^2l^2 + 2\beta l + 2)]\}\mathbb{E}\|\nabla\Phi(\hat{\boldsymbol{\omega}}_t) - \nabla_{\boldsymbol{\omega}} f(\hat{\boldsymbol{\omega}}_t, \boldsymbol{\theta}_t)\|^2$$
$$- \alpha(\frac{\eta_{\boldsymbol{\theta}}}{2} - \frac{l\eta_{\boldsymbol{\theta}}^2}{2})\mathbb{E}\|\nabla_{\boldsymbol{\theta}} f(\hat{\boldsymbol{\omega}}_t, \boldsymbol{\theta}_t)\|^2 + [(1+\alpha)(5\beta^2l^2 + 2)\frac{L\eta_{\boldsymbol{\omega}}^2}{2}$$
$$+ \alpha(\frac{l^2\beta^2\eta_{\boldsymbol{\omega}}}{2} + l\eta_{\boldsymbol{\omega}}^2(5\beta^2l^2 + 2))$$
$$+ \alpha(\frac{l\eta_{\boldsymbol{\theta}}^2}{2} + \frac{l^2\eta_{\boldsymbol{\omega}}^2\eta_{\boldsymbol{\theta}}}{2}(5\beta^2l^2 + 2))]\frac{\sigma^2}{M}$$
$$+ ((1+\alpha)(L + \frac{8}{\eta_{\boldsymbol{\omega}}}) + \alpha(2l + \frac{8}{\eta_{\boldsymbol{\omega}}}) + \alpha l^2\eta_{\boldsymbol{\theta}})d\Delta^2$$

Since we have the following property according to Lemma:

$$\|\nabla\Phi(\hat{\boldsymbol{\omega}}_t) - \nabla_{\boldsymbol{\omega}} f(\hat{\boldsymbol{\omega}}_t, \boldsymbol{\theta}_t)\| \leq l\|\boldsymbol{\theta}^*(\hat{\boldsymbol{\omega}}_t) - \boldsymbol{\theta}_t\|$$
$$\leq \kappa\|\nabla_{\boldsymbol{\theta}} f(\hat{\boldsymbol{\omega}}_t, \boldsymbol{\theta}_t)\|$$

So we can further the above inequality as follows:

$$\mathbb{E}[V_{t+1}] - \mathbb{E}[V_t]$$
$$\leq -\{\frac{\eta_{\boldsymbol{\omega}}}{2}(1+\alpha)(\frac{15}{16} - 5\beta l - 2L\eta_{\boldsymbol{\omega}}(4\beta^2l^2 + 2\beta l + 2))$$
$$- 2\alpha[(\frac{49\eta_{\boldsymbol{\omega}}}{32} + \frac{l^2\beta^2\eta_{\boldsymbol{\omega}}}{2} + l\eta_{\boldsymbol{\omega}}^2(4\beta^2l^2 + 2\beta l + 2))$$
$$+ l^2\eta_{\boldsymbol{\omega}}^2\eta_{\boldsymbol{\theta}}(4\beta^2l^2 + 2\beta l + 2)]\}\mathbb{E}\|\nabla\Phi(\hat{\boldsymbol{\omega}}_t)\|^2$$
$$- \{\alpha(\frac{\eta_{\boldsymbol{\theta}}}{2} - \frac{l\eta_{\boldsymbol{\theta}}^2}{2}) - \kappa^2[(1+\alpha)(\frac{\eta_{\boldsymbol{\omega}}}{2}(1 + \frac{1}{2}\beta l)$$
$$+ L\eta_{\boldsymbol{\omega}}^2(4\beta^2l^2 + 2\beta l + 2)) + 2\alpha[(\frac{49\eta_{\boldsymbol{\omega}}}{32} + \frac{l^2\beta^2\eta_{\boldsymbol{\omega}}}{2}$$
$$+ l\eta_{\boldsymbol{\omega}}^2(4\beta^2l^2 + 2\beta l + 2))$$

$$+ l^2\eta_{\boldsymbol{\omega}}^2\eta_{\boldsymbol{\theta}}(2\beta^2 l^2 + \beta l + 1)]]\}$$

$$\cdot \mathbb{E}\|\nabla_{\boldsymbol{\theta}} f(\hat{\boldsymbol{\omega}}_t, \boldsymbol{\theta}_t)\|^2 + [(1+\alpha)(5\beta^2 l^2 + 2)\frac{L\eta_{\boldsymbol{\omega}}^2}{2}$$

$$+ \alpha(\frac{l^2\beta^2\eta_{\boldsymbol{\omega}}}{2} + l\eta_{\boldsymbol{\omega}}^2(5\beta^2 l^2 + 2))$$

$$+ \alpha(\frac{l\eta_{\boldsymbol{\theta}}^2}{2} + \frac{l^2\eta_{\boldsymbol{\omega}}^2\eta_{\boldsymbol{\theta}}}{2}(5\beta^2 l^2 + 2))]\frac{\sigma^2}{M}$$

$$+ ((1+\alpha)(L+\frac{8}{\eta_{\boldsymbol{\omega}}})+\alpha(2l + \frac{8}{\eta_{\boldsymbol{\omega}}})+\alpha l^2\eta_{\boldsymbol{\theta}})d\Delta^2$$

Then we require the parameters satisfy: $\alpha = \frac{1}{16}$, $\beta l \le \frac{1}{16}$, $\eta_{\boldsymbol{\omega}}(2\beta l + 1)^2\kappa l \le \frac{1}{64}$, $\kappa^2\eta_{\boldsymbol{\omega}} l \le \frac{1}{128}$ and $\eta_{\boldsymbol{\theta}} = 64\kappa^2\eta_{\boldsymbol{\omega}}$

So the inequality can be further simplified as:

$$\mathbb{E}[V_{t+1}] - \mathbb{E}[V_t]$$

$$\le -\frac{\eta_{\boldsymbol{\omega}}}{16}\mathbb{E}\|\nabla\Phi(\hat{\boldsymbol{\omega}}_t)\|^2 - \frac{7}{32}\eta_{\boldsymbol{\omega}}\kappa^2\mathbb{E}\|\nabla_{\boldsymbol{\theta}} f(\hat{\boldsymbol{\omega}}_t, \boldsymbol{\theta}_t)\|^2$$

$$+ \frac{\eta_{\boldsymbol{\omega}}\kappa^4}{64}\frac{\sigma^2}{M} + (\frac{l^2\eta_{\boldsymbol{\theta}}}{16} + \frac{17}{\eta_{\boldsymbol{\omega}}})d\Delta^2$$

$$(39)$$

Telescoping the above inequality we can get:

$$\frac{1}{T}\sum_{t=0}^{T-1}\mathbb{E}\|\nabla\Phi(\hat{\boldsymbol{\omega}}_t)\|^2$$

$$\le \mathcal{O}(\frac{\Phi(\hat{\boldsymbol{\omega}}_0) - \Phi^*}{\eta_{\boldsymbol{\omega}} T})+\mathcal{O}(\frac{\eta_{\boldsymbol{\omega}}\kappa^4\sigma^2}{M})+\mathcal{O}((\eta_{\boldsymbol{\theta}}+\frac{1}{\eta_{\boldsymbol{\omega}}})d\Delta^2)$$

$$(40)$$

$$\square$$