# OpenReview forum: "Zero-shot Sharpness-Aware Quantization for Pre-trained Language Models"
_EMNLP/2023/Conference — EMNLP 2023 Main_

### Official Review · Reviewer_vWQy · 2023-08-02

**Soundness:** 3

**Excitement:**

3: Ambivalent: It has merits (e.g., it reports state-of-the-art results, the idea is nice), but there are key weaknesses (e.g., it describes incremental work), and it can significantly benefit from another round of revision. However, I won't object to accepting it if my co-reviewers champion it.

**Paper Topic And Main Contributions:**

This work proposes a zero-shot quantization framework for textual PLM, which makes Zero-shot Adversarial Quantization (https://arxiv.org/abs/2103.15263) practical for NLP and boosted the generalization of quantized model with a proposed Sharpness-Aware Minimization variant (https://arxiv.org/abs/2010.01412), namely SAM-SGA. Besides, the authors provide the theoretical convergence analysis for this adversarial optimization problem. The experimental results on 11 tasks with generative and discriminative PLM empirically demonstrates the effectiveness of this framework, especially for low-bit weights.


**Questions For The Authors:**

1. The novelty and technique improvement of this submission may be limited. It resorts to Zero-shot Adversarial Quantization (https://arxiv.org/abs/2103.15263) and Sharpness-Aware Minimization variant (https://arxiv.org/abs/2010.01412) to tackle the zero-shot quantization of NLP, and both of them are not the main contributions of this submission.
2. The results of ZeroQuant (Baseline) in Table 1 and 2 is much lower than that in the original paper, i.e., original W8A8-Avg 84.07 v.s. W8A8-Avg 81.78, which is not mentioned in this submission. The author should provide more information about this. From my perspective, this may be caused by the quantization of multi-head self-attention’s weights and the finetuning, but the author should prove it or provide some other explanations.
3. The ablation results on the full GLUE should be provided in section 5.3. For example, in Table 4, the ablation results on MRPC indicate that SGA has negative effects, which does not support the authors’ claim “without the SGA, our SAM-SGA would perform much worse” in line 490. Besides, according to Table 4, SGA seems much more important than SAM and the authors do not mention it.
4. The Task Generalization and Visualization of Landscape experiment do not indicate that the proposed framework leads to better generalization. The better generalization may resort to the SAM which is not the contribution of this submission. Hence, the claim “These results prove that our SAM-SGA can smooth the loss landscape and improve the generalization of PLMs effectively.” should be reconsidered.

**Reasons To Accept:**

- The zero-shot quantization is a practical research problem for textual PLM and even LLM. This work may be one of the first attempts to transfer Zero-shot Adversarial Quantization from CV to NLP. The theoretical convergence analysis and extensive experimental results indicate its effectiveness.


**Reasons To Reject:**

1. The novelty and technique improvement of this submission may be limited. It resorts to Zero-shot Adversarial Quantization (https://arxiv.org/abs/2103.15263) and Sharpness-Aware Minimization variant (https://arxiv.org/abs/2010.01412) to tackle the zero-shot quantization of NLP, and both of them are not the main contributions of this submission.
2. The results of ZeroQuant (Baseline) in Table 1 and 2 is much lower than that in the original paper, i.e., original W8A8-Avg 84.07 v.s. W8A8-Avg 81.78, which is not mentioned in this submission. The author should provide more information about this. From my perspective, this may be caused by the quantization of multi-head self-attention’s weights and the finetuning, but the author should prove it or provide some other explanations.
3. The ablation results on the full GLUE should be provided in section 5.3. For example, in Table 4, the ablation results on MRPC indicate that SGA has negative effects, which does not support the authors’ claim “without the SGA, our SAM-SGA would perform much worse” in line 490. Besides, according to Table 4, SGA seems much more important than SAM and the authors do not mention it.
4. The Task Generalization and Visualization of Landscape experiment do not indicate that the proposed framework leads to better generalization. The better generalization may resort to the SAM which is not the contribution of this submission. Hence, the claim “These results prove that our SAM-SGA can smooth the loss landscape and improve the generalization of PLMs effectively.” should be reconsidered.

**Reproducibility:**

4: Could mostly reproduce the results, but there may be some variation because of sample variance or minor variations in their interpretation of the protocol or method.

**Reviewer Confidence:**

4: Quite sure. I tried to check the important points carefully. It's unlikely, though conceivable, that I missed something that should affect my ratings.

---

> ### Author Rebuttal · Authors · 2023-08-28
>
> # Response to Reviewer #vWQy:
>
> *Thank you for your careful evaluation and constructive feedback! We have addressed all your questions and concerns in the following response. You can refer to the detailed responses for further clarification and verification.*
>
> ***Q1: "Limited novelty and technical improvement."***
> We would like to clarify that our proposed ZSAQ method is **not a straightforward combination** of the two references you mentioned.
>
> 1) First, it is **nontrivial to apply zero-shot adversarial quantization in the NLP applications**, as backpropagation on discrete words is not reasonable. To the best of our knowledge, we are the (nearly) the first to migrate generative-adversarial quantization techniques in computer vision to the NLP domain, which is also well recognized by Reviewer #FTK8 (commented "For the first time in this domain").
>
> 2) Second, different from simply adopting the SAM technique, we **provide a rigorous analysis** for our SAM-SGA and theoretically establish a convergence rate of $\mathcal{O}(1/\sqrt{T})$ for both nonconvex-strongly-concave and nonconvex-PL cases, shedding light on the method's effectiveness even under challenging conditions. The contribution of theoretical analysis is well recognized by Reviewer #2wRB and #FTK8.
>
>
> ---
>
>
> ***Q2: "Explanations for the results that ZeroQuant (Baseline) in Table 1 and 2 is much lower than that in the original paper."***
> Since the settings (e.g., random seed, hyper-parameters) for fine-tuning the full-precision models might be different, the final results of ZeroQuant are different from those of the original paper. However, it should be emphasized that we **implement the ZeroQuant using the official codebase**. We will clarify it in the revision to make the results more convincing.
>
>
> ---
>
>
> ***Q3: "The ablation results on the full GLUE should be provided in section 5.3."***
> Following your suggestions, we provide all the ablation experiment results of full GLUE datasets in the following table. As seen, compared to the full SGA-SAM, the other variants consistently perform worse in most tasks, which can prove the importance and effectiveness of all components.
>
> | Method         |  CoLA |  MNLI |  MRPC |  QNLI |  QQP  |  RTE  |  SST2 |  STSB |  Avg. |
> |----------------|:-----:|:-----:|:-----:|:-----:|:-----:|:-----:|:-----:|:-------:|:-----|
> | Baseline       | 52.13 | 77.24 | 82.60 | 85.10 | 86.34 | 50.90 | 89.11 | 84.41 | 75.98 |
> | SGA-SAM (Ours) | 53.01 | 76.64 | 83.58 | 86.12 | 86.16 | 64.26 | 89.68 | 86.94 | 78.30 |
> | -w/o SGD       | 45.30 | 78.06 | 84.31 | 85.53 | 86.05 | 55.23 | 89.36 | 86.59 | 76.30 ($\downarrow$2.00) |
> | -w/o SAM       | 50.74 | 76.29 | 82.60 | 85.74 | 85.93 | 63.53 | 89.44 | 86.46 | 77.59 ($\downarrow$0.71) |
> | -w/o SGA & SAM | 43.70 | 76.02 | 82.80 | 85.33 | 85.56 | 56.04 | 89.48 | 86.11 | 75.63 ($\downarrow$2.67) |
>
> ---
>
> ***Q4: "The claim “These results prove that our SAM-SGA can smooth the loss landscape and improve the generalization of PLMs effectively.” should be reconsidered."***
> As stated in the Introduction, minimizing the difference between the teacher and (quantized) student models would be prone to over-fitting problems, leading to poor model generalization. Hence, we are inspired by the SAM and propose to **incorporate it as a key component in our SAM-SGA**. Although the SAM itself is not the contribution of our paper, **using it to tackle the challenge of zero-shot adversarial quantization and providing a theoretical analysis are our contributions**. We will clarify it in our revision.

---

### Official Review · Reviewer_FTK8 · 2023-08-06

**Soundness:** 3

**Excitement:**

3: Ambivalent: It has merits (e.g., it reports state-of-the-art results, the idea is nice), but there are key weaknesses (e.g., it describes incremental work), and it can significantly benefit from another round of revision. However, I won't object to accepting it if my co-reviewers champion it.

**Paper Topic And Main Contributions:**

Quantization is a promising approach for reducing memory overhead and accelerating inference. This paper quantizes pre-trained language models without training data, aiming to achieve zero-shot quantization without much loss of accuracy. This is important to protect privacy and ensure data security.

The authors propose a novel zero-shot sharpness-aware quantization (ZSAQ) framework for the zero-shot quantization of various PLMs. The main method is to improve the quantization accuracy and improve the model generalization ability by optimizing the minimax problem. At the same time, the authors provide a theoretical convergence guarantee for the SAM-SGA algorithm which aims to solve the minimax optimization problem (ZSAQ).

I think ZSAQ has the potential to scale to larger language models, and the paper demonstrates that SAM-SGA does indeed lead to better model generalization.

**Questions For The Authors:**

The authors can expand the scope of the experiments to involve more tasks and datasets to verify the generality and effect of the ZSAQ method in different domains.

**Reasons To Accept:**

1. The authors migrate generative adversarial techniques in computer vision to this domain. For the first time in this domain, the authors employ adversarial learning methods to quantify discriminative and generative PLMS in a zero-shot manner and provide rigorous convergence analysis.

2. The ZSAQ method achieves consistent and significant performance gains on both discriminative and generative language models across different tasks, with an average score improvement of 6.98 units over the baseline method.

3. In the paper, the author proved that SAM-SGA algorithm can improve the generalization ability of the model, and verified it on the model of billion-level large models. I think this is very worthy of reference for training large models under insufficient resources.


**Reasons To Reject:**

This paper does not fully discuss and compare other existing zero-shot quantization methods, especially for NLP applications. A comprehensive comparative analysis with these methods can better assess the strengths of ZSAQ.

**Reproducibility:**

3: Could reproduce the results with some difficulty. The settings of parameters are underspecified or subjectively determined; the training/evaluation data are not widely available.

**Reviewer Confidence:**

3: Pretty sure, but there's a chance I missed something. Although I have a good feel for this area in general, I did not carefully check the paper's details, e.g., the math, experimental design, or novelty.

---

> ### Author Rebuttal · Authors · 2023-08-28
>
> # Response to Reviewer #FTK8:
>
> *Thank you for your careful evaluation and constructive feedback! We have addressed all your questions and concerns in the following response. You can refer to the detailed responses for further clarification and verification.*
>
>
> ***Q1: "Not enough comparisons with other zero-shot quantization methods."***
> Following your suggestions, we compare our method with more zero-shot quantization methods, i.e., PTQ-vanilla (vanilla post-training quantization), PTQ-PEG *[Ref 1]* and PTQ-Adaround *[Ref 2]*. Specifically, taking the BERT-base as an example, we show the contrastive results in the W8A8 and W4A8 settings as follows. As seen, our method SAM-SGA **outperforms** the other zero-shot quantization methods **by a clear margin**, especially in the low-bit setting. We will add more contrastive results in our revised paper.
> | Method         | #Bits (W-A) |  CoLA |  MNLI |  MRPC |  QNLI |  QQP  |  RTE  |  SST2 |  STSB |  Avg. |
> |----------------|:-----------:|:-----:|:-----:|:-----:|:-----:|:-----:|:-----:|:-----:|:-----:|:-----:|
> | full-precision |    W32A32   | 60.82 | 83.12 | 85.05 | 90.52 | 89.91 | 65.34 | 92.43 | 88.84 | 82.00 |
> | PTQ-vanilla    |     W8A8    | 58.29 | 82.08 | 80.64 | 88.72 | 87.47 | 62.82 | 90.37 | 88.10 | 79.81 |
> | PTQ-PEG *[Ref 1]*       |     W8A8    | 57.04 | 83.15 | 85.29 | 89.95 | 89.36 | 64.62 | 90.71 | 88.34 | 81.06 |
> | PTQ-Adaround *[Ref 2]*  |     W8A8    | 61.32 | 83.68 | 84.07 | 90.43 | 89.84 | 66.06 | 92.32 | 88.45 | 82.02 |
> | **SAM-SGA (Ours)** |     W8A8    | 63.70 | 82.93 | 85.05 | 90.52 | 89.93 | 67.51 | 92.08 | 88.96 | **82.59** |
> | PTQ-vanilla    |     W4A8    |  1.88 | 33.91 | 31.62 | 50.06 | 62.97 | 52.71 | 48.62 | -1.23 | 35.07 |
> | PTQ-PEG  *[Ref 1]*      |     W4A8    |  0.00 | 37.91 | 31.62 | 50.23 | 63.18 | 47.29 | 73.62 | 11.83 | 39.46 |
> | PTQ-Adaround *[Ref 2]*  |     W4A8    | 55.67 | 72.59 | 79.90 | 87.13 | 86.49 | 64.62 | 91.28 | 82.21 | 77.49 |
> | **SAM-SGA (Ours)** |     W8A8    | 53.01 | 76.64 | 83.58 | 86.12 | 86.16 | 64.26 | 89.68 | 86.94 | **78.30** |
>
> *[Ref 1] Understanding and Overcoming the Challenges of Efficient Transformer Quantization*
>
> *[Ref 2] Up or Down? Adaptive Rounding for Post-Training Quantization*
>
> ---
>
> ***Q2: "Expand the scope of the experiments to involve more tasks and datasets to verify the generality and effect of the ZSAQ method in different domains."***
> We further conduct experiments on more tasks and datasets for OPT-350m and present the results in the following table. It can be seen that our method SAM-SGA outperforms the other methods, which verifies the generality and effectiveness of our approach.
>
> | Method         | #Bits (W-A) | CoLA  | MNLI  | MRPC  | QNLI  | QQP   | RTE   | SST2  | STSB  | Avg. |
> |----------------|:-------------:|:-------:|:-------:|:-------:|:-------:|:-------:|:-------:|:-------:|:-------:|:---------:|
> | full-precision |    W32A32   | 57.51 | 84.54 | 83.33 | 90.30 | 90.72 | 69.68 | 93.23 | 88.13 |  82.18  |
> | Baseline       |     W4A8    | 55.69 | 84.03 | 82.60 | 89.73 | 90.25 | 67.15 | 92.77 | 87.93 |  81.27  |
> | QAT-GT         |     W4A8    | 55.94 | 83.85 | 83.58 | 89.85 | 90.32 | 68.23 | 93.12 | 88.03 |  81.62  |
> | QAT-Rand       |     W4A8    | 55.71 | 83.75 | 82.60 | 89.77 | 90.17 | 67.51 | 92.78 | 88.03 |  81.29  |
> | QAT-Rand-Auto. |     W4A8    | 55.37 | 83.69 | 83.09 | 89.96 | 90.27 | 67.51 | 93.12 | 87.67 |  81.34  |
> | **SAM-SGA**        |     W4A8    | 56.47 | 83.94 | 83.33 | 89.96 | 90.27 | 67.87 | 93.23 | 88.03 |  **81.64 (+0.37)**  |
>
> Note that "QAT-Rand-Auto." denotes a more powerful QAT-based method, as suggested by reviewer #2wRB.

---

### Official Review · Reviewer_2wRB · 2023-08-10

**Typos Grammar Style And Presentation Improvements:** 447, "will be much slight" should be …
**Soundness:** 3

**Excitement:**

3: Ambivalent: It has merits (e.g., it reports state-of-the-art results, the idea is nice), but there are key weaknesses (e.g., it describes incremental work), and it can significantly benefit from another round of revision. However, I won't object to accepting it if my co-reviewers champion it.

**Paper Topic And Main Contributions:**

The paper proposes a method to improve the accuracy of quantized transformer models after QAT (which itself is placed after first floating-point training). The method modifies the loss functions of QAT to include a loss landscape-aware term (namely the sharpness-aware) that encourages the learned model to be more robust to epsilon-perturbations. Experiments on BERT and OPT models demonstrate the efficacy of the method.

**Questions For The Authors:**

Are the adversarial training stages visible on the loss curve? How much effort is required to battle the instability often present in adversarial training? Do the hyperparameters like \eta require careful tweaking for best performance?

**Reasons To Accept:**

The proposed method deals with the important problem of reducing weights of transformers to below 8-bit. Theoretical analysis has been applied to gauge the convergence rate of the proposed method.

**Reasons To Reject:**

The motivation for zero-shot quantization for generative tasks (like the OPT model on wikitext2) is not convincing. Since generative models can start from virtually any token in the embedding matrix and fill-in the rest with the autoregressive generation, there should not be a lack of fine-tuning dataset for them. As by table 3, QAT-GT is much more advantageous than QAT-Rand and the proposed method for Wikitext, hence one would expect the procedure described above (start from seeds in embedding matrix and autoregressively generate the rest) would server as a more suitable and stronger baseline.

The method has multiple stages. It is made of three stages: first training with floating point to get a pretrained model, then do PTQ on the pretrained model with range-shaping techniques (SmoothQuant and GPTQ) to get the initial quantized model, then a QAT fine-tuning process is introduced to recover accuracy loss encountered in the PTQ process. As this method involves many components and stages, it is difficult to analyze and get further improvements.




**Reproducibility:**

3: Could reproduce the results with some difficulty. The settings of parameters are underspecified or subjectively determined; the training/evaluation data are not widely available.

**Reviewer Confidence:**

4: Quite sure. I tried to check the important points carefully. It's unlikely, though conceivable, that I missed something that should affect my ratings.

---

> ### Author Rebuttal · Authors · 2023-08-28
>
> # Response to Reviewer #2wRB:
>
> *Thank you for your careful evaluation and constructive feedback! We have addressed all your questions and concerns in the following response. You can refer to the detailed responses for further clarification and verification.*
>
>
> ***Q1: "Not convincing motivation for zero-shot quantization for generative tasks (like the OPT model on wikitext2)."***
> We respectfully remind you that there exist some quantization scenarios, such as quantizing the financial and medical chatbots, which **rely the task-specific/domain-specific data that are difficult to collect**. Hence, we believe it is critical to explore the zero-shot quantization approach for generative models/tasks. We will clarify it to make our motivation more convincing in the revision.
>
> ---
>
>
> ***Q2: "A more suitable baseline in Table 3."***
> Following your suggestions, we implement this powerful QAT-based baseline (denoted as QAT-Rand-Auto.) and add its results in the following table. As seen, compared to the QAT-Rand, the added "QAT-Rand-Auto." indeed achieves better performance, proving its effectiveness.
> |     Method     | #Bits (W-A) | PTB   | WikiText103 | WikiText2 |
> |--------------|:-------------:|:-------:|:-------------:|:-----------:|
> |                | (W-A)       | PPL   | PPL         | PPL       |
> | full-precision |    W32A32   | 28.58 |    13.38    |   19.21   |
> | Baseline       |     W8A8    | 56.02 |    48.16    |    55.4   |
> | QAT-GT         |     W8A8    | 50.68 |    42.15    |   46.29   |
> | QAT-Rand       |     W8A8    | 54.59 |    48.69    |   60.46   |
> | "QAT-Rand-Auto." |     W8A8    | 52.97 |    47.38    |   53.38   |
> | **SAM-SGA**        |     W8A8    | 46.59 |    46.56    |   54.08   |
> | Baseline       |     W4A4    | 60.25 |    56.16    |   62.89   |
> | QAT-GT         |     W4A4    | 55.59 |    42.01    |   51.72   |
> | QAT-Rand       |     W4A4    |  59.4 |    55.32    |   63.63   |
> | "QAT-Rand-Auto." |     W4A4    | 57.34 |    55.13    |   61.68   |
> | **SAM-SGA**        |     W4A4    | 51.19 |    54.32    |   61.22   |
>
> However, we should note that our **SAM-SGA can still outperform it in many tasks**. And the results in the response of Q2 (Reviewer #6MGP) also show our method brings remarkable performance improvement in other OPT-based tasks. These results prove the superiority of our method.
>
> ---
>
> ***Q3: "It is difficult to analyze and get further improvements due to this method involves many components and stages."***
> Actually, we have conducted a detailed ablation study in Section 5.3 to investigate the effect of all important components and stages. These results **consistently prove the effectiveness and importance of all components**.
>
> For example, we have compared SAM-SGA with *w/o SGA*, *w/o SAM*, and *w/o SGA&SAM* respectively to identify the influence of each stage. And our results in Table 4 (more detailed results can be seen in the response of Q7, Reviewer #6MGP) indicate that removing any component will cause performance degradation and our method will perform much worse without SGA, showing the importance of the adversarial training stage.
>
> ---
>
> ***Q4: "Are the adversarial training stages visible on the loss curve?"***
> Due to the lack of access to submit a PDF file, we can hardly provide the visualization of the loss curve. Instead, we provide the table that documents the variation trend of student_loss and generator_loss respectively. We can see the student_loss decreasing and generator_loss increasing as time varies and eventually reaching a relatively stable plateau during the final stage.
>
> | training loss on CoLA |              |                | 　 | training loss on MNLI |              |                |
> |:---------------------:|:------------:|:--------------:|----|:---------------------:|:------------:|:--------------:|
> |          step         | student_loss | generator_loss |    |          step         | student_loss | generator_loss |
> |           5           |     0.377    |      0.880     |    |           5           | 0.321        | 1.014          |
> |           10          |     0.327    |      1.214     |    |           10          | 0.301        | 1.183          |
> |           15          |     0.341    |      1.449     |    |           15          | 0.220        | 1.407          |
> |           20          |     0.323    |      1.599     |    |           20          | 0.183        | 1.575          |
> |           25          |     0.308    |      1.705     |    |           25          | 0.160        | 1.690          |
> |           30          |     0.295    |      1.784     |    |           30          | 0.149        | 1.764          |
> |           35          |     0.279    |      1.844     |    |           35          | 0.145        | 1.815          |
> |           40          |     0.264    |      1.892     |    |           40          | 0.135        | 1.846          |
> |           45          |     0.249    |      1.927     |    |           45          | 0.129        | 1.867          |
> |           50          |     0.244    |      1.950     | 　 |           50          | 0.118        | 1.878          |
>
>
> ---
>
> ***Q5: "How much effort is required to battle the instability often present in adversarial training? Do the hyperparameters like $\eta$ require careful tweaking for best performance?"***
> Although adversarial training usually suffers from instability, it is relatively stable in our SAM-SGA framework.
>
> First, we theoretically prove the convergence rate under the constraints that $\eta_{\theta}\leq64\kappa^2\eta_{\omega}$ and $\eta_{\omega}\leq\min\\{\frac{1}{64\kappa l(2\beta l+1)^2},\frac{1}{128\kappa^2l}\\}$.
>
> Second, we empirically conduct a grid search for generator/student learning rates ($\eta_{\theta}$/$\eta_{\omega}$) and we find that the results do not exhibit significant fluctuations while demonstrating stability within a narrow range. These prove the effectiveness of our method.
>
> | BERT-base,W4A8,MNLI |       |       |       |       |
> |:-------------------------:|:-------:|:-------:|:-------:|:-------:|
> | $\eta_{\theta}$ \ $\eta_{\omega}$ | 5E-06 | 1E-05 | 2E-05 | 5E-05 |
> | 5E-06                   | 76.23 | 75.84 | 75.90 | 76.50 |
> | 1E-05                   | 76.49 | 76.14 | 74.99 | 74.99 |
> | 2E-05                   | 76.47 | 75.95 | 76.56 | 74.38 |
> | 5E-05                   | 75.96 | 76.64 | 76.04 | 75.72 |
>
> | BERT-base,W4A8,QNLI |       |       |       |       |
> |:-------------------------:|:-------:|:-------:|:-------:|:-------:|
> | $\eta_{\theta}$ \ $\eta_{\omega}$ | 5E-06 | 1E-05 | 2E-05 | 5E-05 |
> | 5E-06                   | 84.46 | 83.76 | 85.00 | 86.12 |
> | 1E-05                   | 85.45 | 84.68 | 83.01 | 84.70 |
> | 2E-05                   | 84.90 | 85.70 | 85.21 | 83.56 |
> | 5E-05                   | 85.65 | 84.57 | 84.04 | 82.92 |
>
>
> ---
>
> ***Typos: 447, "will be much slight" should be "will be much slighter" 466, "universality our method" should be "universality of our method".***
> We apologize for our oversight in paper writing and appreciate your careful reading. We will revise these typos in our paper and polish our writing.

---

### Official Review · Reviewer_6MGP · 2023-08-11

**Soundness:** 4

**Excitement:**

4: Strong: This paper deepens the understanding of some phenomenon or lowers the barriers to an existing research direction.

**Paper Topic And Main Contributions:**

The paper tackles two issues related to LM quantization: access to original training data for Quantization aware training and loss of generality for quantized models due to sharper loss landscapes. From the earlier works referenced by this paper, it seems that the concept of sharpness aware quantization is more prevalent in computer vision models and underexplored for LMs. The paper also provides an analysis of the sharpness-aware-minimization algorithm in an adversarial min-max optimization setting.

The solution to the first problem is provided by introducing an adversarial training setup, where a generator generates synthetic data for both the non-quantized teacher model and the quantized student model. The generator's goal is to produce data which maximizes the discrepancy between the teacher and the student model. The goal of the student model (which is being undergoing quantization aware training) is to minimize this discrepancy.

The solution to the generality problem is tackled by introducing a loss-landscape sharpness aware term in the min-max adversarial loss setup for quantization aware training using a generator.

**Questions For The Authors:**

- the data on which the generators were trained on should me more clear in the experiments section

- Table 2 (BERT large experiments) and 4 (ablating components of SAM-SGA) need more rigorous experiments; given my experience with RTE and MRPC, these tasks are notoriously random in nature (RTE performance varies by a lot for different seeds while finetuning or while performing in-context learning; MRPC task is highly unbalanced and can get high performance by sending all positive labels)

**Reasons To Accept:**

- the tackles an important problem of access to the original pretraining data while performing quantization aware training.
- introduces an adversarial setup training setup which for quantization aware training in conjunction with sharpness aware minimization loss
- adversarial training setup seems to benefit BERT models at lower precision weight quantization than simple ground truth-based quantization-aware-training
- paper shows that sharpness-aware-minimization works in an adversarial-setup for language models

**Reasons To Reject:**

- the claim that the adversarial setup works well for both BERT and generative style models is a slight overclaim
- the benefits of adversarial training on perplexity numbers for OPT-350M in Table 3 are inconclusive
- both OPT-350M and OPT-1.3B have a large gap between full-precision and quantized setups
- although the paper wants to solve the access to data problem using an adversarial setting, it is unclear if the generators used are themselves trained on end-task data
- the benefit of adversarial training is more for lower precision end-task results on BERT (as seen for W2A8 in Table 1), rather than solving the access to the pretraining data problem for quantization-aware-training


**Reproducibility:**

4: Could mostly reproduce the results, but there may be some variation because of sample variance or minor variations in their interpretation of the protocol or method.

**Reviewer Confidence:**

4: Quite sure. I tried to check the important points carefully. It's unlikely, though conceivable, that I missed something that should affect my ratings.

---

> ### Author Rebuttal · Authors · 2023-08-28
>
> # Response to Reviewer #6MGP:
>
> *Thank you for your thoughtful and positive comments! We have addressed all your questions and concerns in the following response. You can refer to the detailed responses for further clarification and verification.*
>
> ***Q1: "It is a slight overclaim that the adversarial setup works well for both BERT and generative style models."***
> It appears that you may be referring to the comparatively lower performance observed in the generative style models. We address this concern in the next questions which you can refer to.
>
> ---
>
> ***Q2: "The benefits of adversarial training on perplexity numbers for OPT-350M in Table 3 are inconclusive."***
> We **expand experiments to more tasks and datasets for OPT-350m**. The contrastive results are listed in the table below, illustrating that our method SAM-SGA outperforms the other methods, which further **validates the effectiveness of our methods for generative style models**.
>
> | Method | #Bits (W-A) | CoLA  | MNLI  | MRPC  | QNLI  | QQP   | RTE   | SST2  | STSB  | Avg. |
> |----------------|:-------------:|:-------:|:-------:|:-------:|:-------:|:-------:|:-------:|:-------:|:-------------------:|:--------------|
> | full-precision |    W32A32   | 57.51 | 84.54 | 83.33 | 90.30 | 90.72 | 69.68 | 93.23 | 88.13 |  82.18  |
> | Baseline       |     W4A8    | 55.69 | 84.03 | 82.60 | 89.73 | 90.25 | 67.15 | 92.77 | 87.93 |  81.27  |
> | QAT-GT         |     W4A8    | 55.94 | 83.85 | 83.58 | 89.85 | 90.32 | 68.23 | 93.12 | 88.03 |  81.62  |
> | QAT-Rand       |     W4A8    | 55.71 | 83.75 | 82.60 | 89.77 | 90.17 | 67.51 | 92.78 | 88.03 |  81.29  |
> | QAT-Rand-Auto. |     W4A8    | 55.37 | 83.69 | 83.09 | 89.96 | 90.27 | 67.51 | 93.12 | 87.67 |  81.34  |
> | **SAM-SGA**        |     W4A8    | 56.47 | 83.94 | 83.33 | 89.96 | 90.27 | 67.87 | 93.23 | 88.03 |  **81.64 (+0.37)**  |
>
> Note that "QAT-Rand-Auto." denotes a more powerful QAT-based method, as suggested by Reviewer #2wRB.
>
> ---
>
>
> ***Q3: "Both OPT-350M and OPT-1.3B have a large gap between full-precision and quantized setups."***
> We have to admit that for generative models, there exists a large gap between full-precision and quantized versions. It's been pointed out that quantization methods for generative style models suffer from homogeneous word embeddings and varied distribution of weights *[Ref 1]*.
> Nonetheless, compared to the other zero-shot quantization methods, our SAM-SGA brings a remarkable performance improvement. This can prove the effectiveness of our method.
>
> *[Ref 1] Compression of Generative Pre-trained Language Models via Quantization*
>
> ---
>
> ***Q4: "It is unclear if the generators used are themselves trained on end-task data."***
> Actually, we directly use the third-party GPT-style PLMs (**without any further tuning**) as the generators, i.e., the generators **are not trained on any end-task data**.  Owing to their powerful generative abilities, we can easily obtain the generated sentences by prompting the generators with some simple instructions, such as "Randomly generate some complete and fluent sentences:". We will clarify it in our revision.
>
> ---
>
> ***Q5: "The benefit of adversarial training is more for lower precision end-task results on BERT, rather than solving the access to the pretraining data problem for quantization-aware-training."***
> Indeed, the primary focus of our work is to address the access to task-specific/domain-specific data problems using an adversarial setting in the context of quantization-aware training.
> While it is notable that ZSAQ performs better for lower precision end-task, which indicates the effectiveness of our proposed approach. And we will emphasize the benefits in lower precision end-task results in the revision.
>
> ---
>
> ***Q6: "The data on which the generators were trained on should be more clear in the experiments section."***
> As mentioned in Q4, we **do not train** the generators with any end-task data.
>
> ---
>
> ***Q7: "Table 2 (BERT large experiments) and 4 (ablating components of SAM-SGA) need more rigorous experiments."***
> 1.  For Table2, following your suggestions, we show the full results of the GLUE benchmark and extend the table below. As seen, our method achieves consistent and significant performance improvements compared to the baseline on BERT-large, confirming the statement of our paper.
> | Method         | #Bits (W-A) |  CoLA |  MNLI |  MRPC |  QNLI |  QQP  |  RTE  |  SST2 |  STSB |  Avg. |
> |----------------|:-----------:|:-----:|:-----:|:-----:|:-----:|:-----:|:-----:|:-----:|:-----------:|:-----|
> | full-precision |    W32A32   | 64.47 | 83.12 | 86.03 | 92.37 | 90.85 | 65.71 | 93.35 | 89.22 | 83.14 |
> | Baseline       |     W8A8    | 64.80 | 83.33 | 86.76 | 92.27 | 90.79 | 62.45 | 93.46 | 89.12 | 82.87 |
> | QAT-GT         |     W8A8    | 66.61 | 83.29 | 87.75 | 92.54 | 90.81 | 65.34 | 93.23 | 89.26 | 83.60 |
> | QAT-Rand       |     W8A8    | 66.86 | 83.01 | 86.52 | 92.42 | 90.76 | 65.15 | 93.00 | 89.18 | 83.36 |
> | **SAM-SGA**        |     W8A8    | 66.08 | 83.50 | 87.25 | 92.55 | 90.87 | 66.06 | 94.04 | 89.37 | **83.72 (+0.85)** |
> | Baseline       |     W4A8    | 49.11 | 78.86 | 74.26 | 86.86 | 86.68 | 47.29 | 87.73 | 87.63 | 74.80 |
> | QAT-GT         |     W4A8    | 50.85 | 79.17 | 75.73 | 90.44 | 87.35 | 56.68 | 92.32 | 88.05 | 77.57 |
> | QAT-Rand       |     W4A8    | 45.63 | 79.02 | 70.34 | 87.26 | 87.17 | 47.29 | 78.84 | 88.08 | 72.95 |
> | **SAM-SGA**        |     W4A8    | 52.97 | 79.17 | 77.70 | 90.69 | 87.20 | 52.71 | 90.48 | 87.72 | **77.33 (+2.53)** |
>
> 2. Similarly, we also report more ablation results in Table 4 as follows. Compared to the full SGA-SAM, the other variants consistently perform worse in most tasks, which can prove the importance and effectiveness of all components.
> | Method         |  CoLA |  MNLI |  MRPC |  QNLI |  QQP  |  RTE  |  SST2 |  STSB |  Avg. |
> |----------------|:-----:|:-----:|:-----:|:-----:|:-----:|:-----:|:-----:|:--------:|:-----|
> | Baseline       | 52.13 | 77.24 | 82.60 | 85.10 | 86.34 | 50.90 | 89.11 | 84.41 | 75.98 |
> | SGA-SAM (Ours) | 53.01 | 76.64 | 83.58 | 86.12 | 86.16 | 64.26 | 89.68 | 86.94 | 78.30 |
> | -w/o SGD       | 45.30 | 78.06 | 84.31 | 85.53 | 86.05 | 55.23 | 89.36 | 86.59 | 76.30 ($\downarrow$2.00) |
> | -w/o SAM       | 50.74 | 76.29 | 82.60 | 85.74 | 85.93 | 63.53 | 89.44 | 86.46 | 77.59 ($\downarrow$0.71) |
> | -w/o SGA & SAM | 43.70 | 76.02 | 82.80 | 85.33 | 85.56 | 56.04 | 89.48 | 86.11 | 75.63 ($\downarrow$2.67) |

---

### Meta-Review · Area_Chair_8VAx · 2023-09-19

**Recommendation:** 4

**Metareview:**

The manuscript addresses critical challenges in the quantization of language models (LMs), focusing on two primary concerns: access to original training data for Quantization Aware Training (QAT) and the loss of generality due to sharper loss landscapes in quantized models. It introduces an adversarial training setup and incorporates sharpness-aware terms in the min-max loss function for QAT. Below is a synthesis of reviewers' perspectives on the paper's strengths and weaknesses.

Strengths:

1. Novelty and Relevance: The paper addresses important problems in LM quantization and introduces new paradigms like adversarial training and sharpness-aware minimization to the domain. These have been prevalent in computer vision but are largely underexplored for LMs.
2. Theoretical Analysis: The paper provides a robust theoretical framework, particularly through its introduction of a sharpness-aware-minimization algorithm in an adversarial min-max optimization setting. This is backed by convergence rate analysis.
3. Empirical Results: The paper demonstrates the efficacy of its method on established models like BERT and OPT, highlighting how the method outperforms the baseline, particularly in terms of model generalization.
4. Transfer of Techniques: The manuscript successfully adapts techniques like adversarial training from computer vision to the domain of NLP, which is commendable.

Weaknesses:
1. Inconclusive Evidence: Certain empirical results, such as the benefits of adversarial training on perplexity numbers for OPT-350M, are not definitively proven.
2. Complexity of the Method: The method involves multiple stages, including floating-point training, post-training quantization (PTQ), and QAT fine-tuning, making it challenging to analyze and extend.
3. Unclear Motivation for Certain Choices: Some reviewers pointed out that the motivation behind zero-shot quantization for generative tasks, and the specific choice of adversarial setup, are not convincingly argued.

---

### Decision · Program_Chairs · 2023-10-07

**Decision:**

Accept-Main

**Comment:**

The manuscript addresses critical challenges in the quantization of language models (LMs), focusing on two primary concerns: access to original training data for Quantization Aware Training (QAT) and the loss of generality due to sharper loss landscapes in quantized models. It introduces an adversarial training setup and incorporates sharpness-aware terms in the min-max loss function for QAT. Below is a synthesis of reviewers' perspectives on the paper's strengths and weaknesses.

Strengths:

1. Novelty and Relevance: The paper addresses important problems in LM quantization and introduces new paradigms like adversarial training and sharpness-aware minimization to the domain. These have been prevalent in computer vision but are largely underexplored for LMs.
2. Theoretical Analysis: The paper provides a robust theoretical framework, particularly through its introduction of a sharpness-aware-minimization algorithm in an adversarial min-max optimization setting. This is backed by convergence rate analysis.
3. Empirical Results: The paper demonstrates the efficacy of its method on established models like BERT and OPT, highlighting how the method outperforms the baseline, particularly in terms of model generalization.
4. Transfer of Techniques: The manuscript successfully adapts techniques like adversarial training from computer vision to the domain of NLP, which is commendable.

Weaknesses:
1. Inconclusive Evidence: Certain empirical results, such as the benefits of adversarial training on perplexity numbers for OPT-350M, are not definitively proven.
2. Complexity of the Method: The method involves multiple stages, including floating-point training, post-training quantization (PTQ), and QAT fine-tuning, making it challenging to analyze and extend.
3. Unclear Motivation for Certain Choices: Some reviewers pointed out that the motivation behind zero-shot quantization for generative tasks, and the specific choice of adversarial setup, are not convincingly argued.